# Retinal Prostheses: Engineering and Clinical Perspectives for Vision Restoration

**DOI:** 10.3390/s23135782

**Published:** 2023-06-21

**Authors:** Kevin Y. Wu, Mina Mina, Jean-Yves Sahyoun, Ananda Kalevar, Simon D. Tran

**Affiliations:** 1Department of Surgery, Division of Ophthalmology, University of Sherbrooke, Sherbrooke, QC J1G 2E8, Canada; yang.wu@usherbrooke.ca (K.Y.W.);; 2Department of Mechanical and Manufacturing Engineering, University of Calgary, Calgary, AB T2N 1N4, Canada; 3Faculty of Medicine, University of Montreal, Montreal, QC H3T 1J4, Canada; 4Faculty of Dental Medicine and Oral Health Sciences, McGill University, Montreal, QC H3A 1G1, Canada

**Keywords:** flexible electronic devices, implantable electronic devices, flexible sensors, human–machine interface, retinal prosthesis, vision restoration, retinal disease, ophthalmology

## Abstract

A retinal prosthesis, also known as a bionic eye, is a device that can be implanted to partially restore vision in patients with retinal diseases that have resulted in the loss of photoreceptors (e.g., age-related macular degeneration and retinitis pigmentosa). Recently, there have been major breakthroughs in retinal prosthesis technology, with the creation of numerous types of implants, including epiretinal, subretinal, and suprachoroidal sensors. These devices can stimulate the remaining cells in the retina with electric signals to create a visual sensation. A literature review of the pre-clinical and clinical studies published between 2017 and 2023 is conducted. This narrative review delves into the retinal anatomy, physiology, pathology, and principles underlying electronic retinal prostheses. Engineering aspects are explored, including electrode–retina alignment, electrode size and material, charge density, resolution limits, spatial selectivity, and bidirectional closed-loop systems. This article also discusses clinical aspects, focusing on safety, adverse events, visual function, outcomes, and the importance of rehabilitation programs. Moreover, there is ongoing debate over whether implantable retinal devices still offer a promising approach for the treatment of retinal diseases, considering the recent emergence of cell-based and gene-based therapies as well as optogenetics. This review compares retinal prostheses with these alternative therapies, providing a balanced perspective on their advantages and limitations. The recent advancements in retinal prosthesis technology are also outlined, emphasizing progress in engineering and the outlook of retinal prostheses. While acknowledging the challenges and complexities of the technology, this article highlights the significant potential of retinal prostheses for vision restoration in individuals with retinal diseases and calls for continued research and development to refine and enhance their performance, ultimately improving patient outcomes and quality of life.

## 1. Introduction

### 1.1. Overview of Retinal Prostheses

In recent years, the field of retinal prosthetics has gained considerable attention due to its potential to restore some useful vision to individuals suffering from retinal diseases that have resulted in the loss of photoreceptors, such as age-related macular degeneration and retinitis pigmentosa. These conditions can lead to significant vision loss, severely impacting patients’ quality of life and ability to perform daily tasks. Retinal prostheses, also known as bionic eyes, offer a promising solution by stimulating the remaining retinal cells to produce visual sensations.

The development of retinal prosthetic devices has been marked by significant technological advancements, including the introduction of various implant designs such as epiretinal, subretinal, and suprachoroidal sensors. These innovations have expanded the range of possibilities for enhancing patients’ visual acuity and improving the overall functionality of retinal prosthetic devices.

This comprehensive review delves into the pre-clinical and clinical studies conducted between 2017 and 2023, exploring the retinal anatomy, physiology, pathology, and principles underlying electronic retinal prostheses. Engineering aspects such as electrode–retina alignment, electrode size and material, charge density, resolution limits, spatial selectivity, and bidirectional closed-loop systems are examined. This article also discusses clinical aspects, focusing on safety, adverse events, visual function, outcomes, and the importance of rehabilitation programs. Moreover, this review addresses the ongoing debate over the viability of implantable retinal devices in light of emerging cell-based and gene-based therapies as well as optogenetics, comparing retinal prostheses to these alternative therapies and providing a balanced perspective on their advantages and limitations. The recent advancements in retinal prosthesis technology are outlined, emphasizing progress in engineering and the future outlook of retinal prostheses. In particular, the introduction of wireless technology and artificial intelligence to improve the processing and transmission of electrical signals to the retinal is discussed.

### 1.2. Overview of Retinal Structure and Function

The retina, a thin and transparent structure, embryologically originates from the inner and outer layers of the optic cup. It comprises 10 distinct layers, each playing a crucial role in the visual processing pathway. In a cross section, these layers can be identified from the inner to the outer retina as follows (Figure 1).

Nerve fiber layer: This layer contains the axons of retinal ganglion cells that coalesce to form the optic nerve.Ganglion cell layer: This layer is composed of the cell bodies of retinal ganglion cells, which transmit visual information to the brain.Inner plexiform layer: This layer consists of the synapses between bipolar and ganglion cells, facilitating signal processing and integration.Inner nuclear layer: This layer houses the cell bodies of bipolar, horizontal, and amacrine cells, which play essential roles in processing and transmitting visual information.Outer plexiform layer: This layer contains synapses between photoreceptors, bipolar cells, and horizontal cells, allowing for the initial processing of visual signals.Outer nuclear layer: This layer is made up of the cell bodies of rod and cone photoreceptors, which are responsible for capturing and converting light into neural signals.Rod and cone inner and outer segments: These segments are part of the photoreceptor cells, which include rods and cones. The inner segments contain vital cellular components, such as mitochondria, while the outer segments contain stacked discs rich in photopigments, which are essential for absorbing light and initiating the phototransduction cascade to produce neuronal signals.Retinal pigment epithelium (RPE): This is the outermost layer of the retina, located just beneath the photoreceptor cells. The RPE has several crucial functions in visual processing. Its cells absorb stray light, preventing light scatter and enhancing visual acuity. They also play a vital role in recycling photopigments and shuttling nutrients to the photoreceptors. Additionally, they facilitate the transport of metabolic waste products from the photoreceptors to the choroidal blood supply, thereby helping to maintain the health of the photoreceptor cells.

Horizontal cells make synaptic connections with rod spherules and cone pedicles, while bipolar cells are vertically oriented, synapsing with either rod or cone synaptic bodies. Their axons make synaptic contact with ganglion cells and amacrine cells in the inner plexiform layer. The axons of ganglion cells form the nerve fiber layer and later the optic nerve, containing over one million optic nerve fibers [1].

### 1.3. Overview of Retinal Physiology and Pathology

Retinal prostheses aim to restore vision in degenerate eyes by replacing the function of the photoreceptors. In the normal eye, the photoreceptors in the outer layers of the retina contain light-sensitive pigments that trigger the phototransduction cascade, generating neuronal signals upon light stimuli. These signals are processed by a complex network of neurons within the middle layers of the retina before reaching retinal ganglion cells (RGCs) in the inner layers. Axonal processes from RGCs form the optic nerve, transmitting light-evoked neuronal signals to the visual cortex. However, in congenital retinal dystrophies such as retinitis pigmentosa, the photoreceptors in the outer layers are gradually lost, causing progressive visual loss, while the inner retinal layers, including RGCs and bipolar cells, are partially spared [2].

Theoretically, vision restoration could be achieved by creating retinal prostheses that receive and process incoming light, transmitting the information as electrical impulses to the remaining inner retinal layers for visual function. However, the retina’s complex physiology poses significant challenges for retinal prostheses, as devices must replicate intricate retinal processing. Currently, electrical or photovoltaic stimulation is provided in a relatively unspecified manner, simultaneously activating multiple cell types. The challenge lies in developing more selective stimulation to optimize device resolution and patient outcomes.

## 2. Principles of Electronic Retinal Prostheses

At the intersection of science and engineering, the field of visual prosthetics is driven by an ambition–to restore vision to the blind. The epitome of success in the field would be to create vision comparable to that of a healthy retina. Natural vision is made possible through complex, highly orchestrated, temporally synchronized electrical signals fired by retinal ganglion cells [3]. This, then, becomes the goal of retinal prosthesis–to receive a light stimulus, convert it to an electric pulse, and deliver that pulse to the retina through a microelectrode array. Through this electric signal, prostheses aim to stimulate the structures of the retina that remain capable of conveying signals to the optic nerve, which transmits them to the brain for interpretation. 

Mainly, two main mechanisms were described that allow prostheses to provide electrical stimulation to retinal cells [4]. The first mechanism works through external light detection and internal impulse delivery. In this mechanism, a microelectrode array is intraocularly implanted. This electrode array cannot convert light signals to electrical impulses by itself. Thus, it depends on an external camera system, such as one mounted on a set of glasses, to receive an image from the surroundings. That image is then processed (i.e., the image is converted to electrical impulses) and sent to a periocular implant (Figure 2). Finally, the periocular implant sends the electrical signal to the implanted microelectrode via cables. The microelectrode array then conveys it to the retinal cells. An example of one such system is the ARGUS II prosthetic (Figure 2 and Figure 3) [5]. In the second mechanism, a photodiode electrode array that can act directly as the image receiver is implanted. This photodiode electrode is able to directly receive light energy and transduce it to an electrical signal (Figure 4) [6]. The light-induced signal can then be used to stimulate retinal cells. These photodiode-based systems aim to replace the lost photoreceptors of the retina, and they can often fulfill their function without having processing units or external cameras. In some cases, however, external components including external power were used to amplify the light waves and produce suitably strengthened impulses [7]. Generally, prostheses using the second mechanism are activated directly by light and can depend on photovoltaic processes, such as creating a voltage drop, to produce an electric field which can then be used to stimulate target cells [8]. Photodiode systems have multiple advantages. (1) Due to their photovoltaic process, photodiode-based prostheses can avoid the use of implanted cables. Consequently, the surgical implantation tends to be less complex for photodiode-based prostheses than for implants requiring cable connections. (2) These systems can directly transduce light stimuli to electric signals and are, thus, often functional without the use of an external camera. Eliminating the need for an external camera maintains the link between eye movements and visual stimuli [9].

Regardless of the mechanism of action, however–whether the prosthesis has internal non-light-sensitive electrodes or photodiodes–both systems must fulfill common engineering design criteria. Hereafter in this review, the general term microelectrode array (MEA) is used to refer to both types of arrays, since both systems use electrodes to convey electric impulses to retinal cells [10].

## 3. Engineering of Retinal Prostheses

The performance of retinal prostheses is dependent on a few outcomes. Fulfilling these outcomes is important to drive their adoption in the market and by patients. Each of the outcomes for retinal prostheses is closely related to an engineering design challenge, namely, a constraint (Figure 5). This review describes the four main outcomes and their corresponding engineering design constraints.

Firstly, retinal prostheses must provide effective electrical stimulation to the retina to enable formation of light percepts. To achieve this outcome, the microelectrode array of prostheses must directly contact the retina for successful stimulation. Even small separations between the microelectrode array and the retina can reduce a prosthetic’s efficacy. Thus, this review discuses some of the challenges and solutions proposed to ensure that retinal prostheses can meet the design consideration. 

Secondly, it is important for retinal prostheses to produce a high-resolution image that can then be interpreted by the recipient. To attain higher resolution, one main consideration is to minimize the size of the individual microelectrodes on the array such that each electrode can stimulate a single retinal cell and maximize the resolution limit of prostheses. However, there are technical challenges with such miniaturizing of microelectrodes. These challenges are described here, along with some of the solutions proposed to overcome them.

Thirdly, another design consideration of retinal prostheses is their ability to selectively target the desired cells in the retina. Different retinal cells produce different perceptions that directly impact the patient outcomes. As such, this review delves into a description of two techniques that can be used to selectively target retinal cells, namely, (1) current steering, which uses return electrodes, and (2) manipulating the electric stimulation parameters.

Finally, an important design consideration is a prosthetic’s ability to be customizable to every recipient’s individual needs. To achieve such an outcome, prostheses must be able to both convey electrical pulses and be able to record the impact of such stimulation on the retina. Thus, a description of the incorporation of bidirectional retinal prostheses is given. 

It is mainly these design criteria that guide the manufacturing of retinal prostheses and dictate their future market adoption. Thus, each of the criteria is described in detail to allow the reader to understand the challenges, learn about recent advancements, and contemplate the prospects of retinal prostheses.

### 3.1. Electrode–Retina (ER) Topographical Alignment

MEAs directly interface with the retina to deliver electrical stimuli. Signals delivered by MEAs are either directly or indirectly received by the retinal ganglion cells (RGCs). In the case of epiretinal prostheses, the electrical signals are directly received, since the prostheses are in direct contact with the RGCs. In the case of subretinal/suprachoroidal prostheses, however, the signals are first received by cells in the posterior retina, which then convey them to the RGCs through the physiologic pathways. The received signal is then conveyed by the RGCs, through the optic nerve, to the brain, where they are interpreted as images [11]. The transmission of an electric stimulus to retinal neurons requires a close topographical fit between the MEA of the retinal prostheses and the retinal tissue [12]. The lack of conformity between implants and retinal tissue can cause gaps up to several hundred micrometers, which result in an impaired or lost signal in these areas and, consequently, impact the effectiveness of the device (Figure 6, A) [13]. Thus, researchers proposed a variety of solutions to overcome this topographical misalignment that can occur between the retina and the implant. 

Firstly, many researchers developed MEAs that incorporate three-dimensional (3D) geometries. Within the retina, experiments showed that cells of the inner neuron layer (INL) can migrate and integrate with 3D geometry MEAs (Figure 6, B) [14]. Since subretinal prostheses are located in the INL, they are the ones that benefit the most from such integration of cells into the 3D geometry of MEAs. It was shown that both the protruding and recessed 3D geometries of subretinal implants benefit from this retinal migration, which reduces the separation distance between the INL and the implanted electrodes [14,15,16,17,18]. However, while this advantage applies to subretinal prostheses, both epiretinal and suprachoroidal implants were reported to have a gradually increasing ER distance over time. This observed phenomenon was attributable to fibrosis and to evoked inflammatory responses [19]. Since this time-based change in ER distance is particularly evident in epiretinal and suprachoroidal prostheses, these prostheses had more studies monitor this measurement, compared to studies of subretinal implants, in which device trials never measured the ER distance [19]. It is also worth noting that while it might be possible for retinal migration to decrease the ER distance in subretinal implants, epiretinal implants require mechanical pressure to achieve close proximity to the MEA [18]. All in all, due to the benefits of incorporating 3D geometries, these MEA designs were considered for subretinal devices. However, epiretinal and suprachoroidal MEAs rely on other methods that are more effective with their implant location. 

Secondly, to decrease the electrode–retina distance, researchers considered integrating pneumatic cavities that enable dynamic, real-time control of the electrode position. In MEAs, pneumatic cavities can be placed under the electrodes [12], and, by adjusting the pneumatic pressure, it then becomes possible to change the electrode position and reduce the distance to the retinal surface. Like the proposed design of pneumatic systems, hydraulic systems can also be tested for incorporation, especially in epiretinal devices, to improve the topographical alignment. 

Thirdly, researchers developed flexible MEA substrates that can be used to achieve topographical alignment. Beyond their use in brain neuron stimulation, flexible probes were suggested for use in epiretinal electrode arrays [20,21]. Such flexible designs can better fit the surface topographies and can, therefore, permit the creation of larger and higher-density devices, which can extend over the curvature of the retina. These flexible designs were also shown to be safe and effective [22,23]. Finally, the flexibility of these MEAs was considered beneficial in reducing the acute insertion footprint of the electrodes, an outcome often measured by retinal cell viability after the implantation [24]. However, while these MEAs are advantageous in many facets, including that they provide better topographical alignment, it can still be challenging for these microelectrode arrays to fill smaller gaps and sharp corners [12]. 

All in all, as an important engineering consideration, a variety of solutions were suggested to decrease the electrode–retina distance in the different types of retinal prostheses. The advantages and disadvantages of each of the solutions should be weighed, and, based on implant location, the appropriate feature should be incorporated.

### 3.2. Electrode Size and Material, Charge Density, and Resolution Limit

While most clinical results from human-implanted prostheses showed positive outcomes, the visual resolution obtained from prostheses has, nevertheless, remained limited. The recognition of simple objects and facial identification can still be challenging [25]. Similarly, the restoration of visual acuity in patients with retinal prosthesis has been, despite pioneering success in human trials, quite limited. Clinically, to describe visual acuity, the Snellen scale is often used. The scale has a large range, but specific acuities are defined—“normal” vision is defined as 20/20, and an acuity of 20/200 is defined as legally blind. The recent clinical trials of the PRIMA system, which is described in further detail in the succeeding sections, reported the best visual acuities of retinal prostheses yet, in the range of 20/460–20/565 [26]. While these acuities remain within the definition of legal blindness, a closer look reveals that there is a close match between the prosthetic acuity and the fundamental sampling limit set by the pixel size [27]. Quantitatively, it was reported that the visual acuity corresponded to 1.17 ± 0.13 of the pixel size [28]. From this observation, it can be inferred that the pixel size of the microelectrodes in the prosthesis was limiting the stimulation of adjacent points on the retina and, thereby, limiting acuity. This close match between the acuity and the fundamental sampling limit of the prosthetic consequently indicates that smaller pixels may provide higher resolution (Figure 7) [27].

In addition to this clinical observation, physiologically, to restore natural vision, retinal prostheses should ideally stimulate individual retinal neurons [11]. There are more than 1.5 million RGCs in the human retina, of which the largest soma has a diameter of about 30 μm [29]. In comparison, the smallest electrode size of the retinal prostheses that have been applied in clinical trials is the 50 × 50 μm electrode of the Alpha IMS system [30]. Therefore, both given the clinical results of the PRIMA system and to emulate physiological processes, it is logical to develop MEAs with smaller, more densely packed electrodes.

From an engineering perspective, one of the primary challenges with miniaturizing MEAs and decreasing pixel size is the corresponding increase in charge density [31]. With decreasing pixel electrode size, there is less surface area available to transfer charge to the surrounding tissue. However, retinal cells require a minimum amount of charge to meet the activation threshold needed in order to relay visual information [32]. Consequently, to provide a similar electric field penetration depth, a higher electrode charge density per surface area is necessary when using smaller electrodes [33]. Due to the limitations of material properties and safety requirements, it is, nevertheless, challenging to accommodate this increased charge-density requirement [34,35]. Conventional platinum electrodes, widely used for visual prostheses, exceed the charge injection limits for safe stimulation with decreased pixel size [31]. To overcome this challenge, innovations in the field of material science have proposed modifications to retinal prostheses’ MEAs. For instance, researchers manufactured nanocone-shaped platinum–iridium oxide neural microelectrodes to increase the electrode surface area [36]. Others incorporated high-conductivity materials such as graphene, materials with superior charge injection such as polymer/carbon nanotubes, and carbon nanotube-modified gold [31,37,38]. These innovations made it possible to have efficient charge injection and opened the horizon to decreasing the electrode size on MEAs, with a final objective of improving the resolution limit of retinal prostheses [39]. However, with each of the proposed materials, biocompatibility, manufacturability, and batch-to-batch consistency are still important considerations that need to be studied to maximize the benefits while ensuring that the risks and variability are minimized. A more comprehensive discussion concerning the innovation within the field of material sciences, as it relates to retinal prostheses, is presented in Section 7.

### 3.3. Spatial Selectivity

Electrical stimulation by MEAs allows surviving RGCs to depolarize and transmit visual signals to the brain [40]. The outcome achieved by these signals is a phosphene–the basic unit of artificial vision, defined as a subjective “visual percept” experienced by recipients of retinal prostheses. Unsurprisingly, in an ethnographic, qualitative account of how the current implementation of retinal prostheses translates into the perceptual experience of patients, the recipients of retinal prostheses described their vision as distinctly different from that of natural vision and as most closely resembling a “light show” [41]. These manifest differences between artificial and natural vision can be attributed to the coarse pattern of retinal activation caused by the MEA stimulation of many neighboring cells without coordination [40]. However, it is not necessarily the difference from natural vision that is concerning, rather, it is the unreliable and irregularly shaped phosphenes that are produced as a result of such coarse and unselective activation [42]. When an electric field is produced by a stimulating electrode, the shape of the electric field and its strength directly impact the visual percept of the recipient. If the electric field spreads in a lateral direction with distance above the electrode, it can simultaneously activate many retinal cells and cause a loss of spatial selectivity (Figure 8) [25].

In epiretinal devices, MEAs inadvertently activate a bundle of RGC axons in the nerve fiber layer. This bundle activation happens since the axons of RGCs lie closer to the electrodes than the underlying RGC soma that are the intended targets of stimulation [43]. The result of this activation is elongated or arc-like phosphenes that negatively impact the visual quality [44]. Similarly, while subretinal prosthesis are largely spared from the interference by RGC axons, the unwanted excitation from these electrodes can result in highly variable phosphene shapes across electrodes and between subjects [45]. This phenomenon happens due to the activation of many cells and cell types, including tertiary retinal neurons such as amacrine cells [46,47].

If the single electrodes of retinal prostheses can reliably produce an individual, isolated point of light, which the brain can then assemble into objects similar to an electronic scoreboard, the spatial resolution and consequent vision can be drastically improved [48]. However, this is not yet possible due to unselective and imprecise retinal stimulation. The engineering solutions to this challenge have varied, and two predominant methods have been proposed. Firstly, there is the use of return electrodes to localize the electric field and, thereby, selectively stimulate cells. Secondly, there is the manipulation of the electric stimulation parameters, such as the amplitude and frequency, which can selectively target cells that only respond to those specific parameters and not to others.

#### 3.3.1. Return Electrodes for Electric Field Localization and Current Steering

The stimulation of retinal cells is mainly achieved by depolarizing cells in an electric field, instead of direct current injection into cells [25]. By accumulating charge at the synaptic terminals of retinal cells, the membrane depolarization exceeds a threshold, and action potentials can be fired. Confining the electric field and controlling its shape is important to ensure the stimulation selectivity of the desired target cells. To achieve such control, researchers manipulated MEAs to incorporate adjacent return or grounding electrodes. Return electrodes serve as the electrical counterpart to the active electrodes that are implanted in the retina, and, through them, it is possible to limit current spread and produce a well-controlled, directed electric field. A variety of return electrodes were widely investigated for use in retinal prostheses including monopolar, bipolar, tripolar, and hexapolar configurations. As seen in Figure 9, in the monopolar configuration, a single stimulating electrode is used to activate retinal cells. The bipolar configuration has an active electrode with an adjacent return electrode, which allows for current steering and localization. In the tripolar configuration, two electrodes at the opposite sides of the stimulating electrode function as the return electrodes. Finally, in the hexapolar configuration, an active electrode is centrally located and is surrounded by six return electrodes that form a “guard” around the active electrode [25]. In addition to these main designs, more complex configurations were recently explored and proposed for further testing. For instance, a 3D multilayered, concentric bipolar electrode was designed with the purpose of achieving highly focused electric stimulation [49]. Another research group developed a method that enables dynamic electric field confinement, by turning the designated active pixels into transient returns [46]. In addition to these designs, researchers also developed a computational implant simulator that can model electric fields of MEAs [27]. This simulator was able to compute the electric field in the retina generated by thousands of electrodes as well as with various pixel configuration [27]. It is through such computational models as well as through the unique ideation and implementation of return electrodes that it can be possible to confine the electric field and selectively target desired cells [40]. 

However, while current steering is important to ensure that small-sized electrodes can target one cell at a time, it does not overcome the challenge of bundle activation in the nerve fiber layer. Especially in epiretinal electrodes, where the nerve fiber layer more proximally lies to the electrode than to the more distal RGC somas, another method of cell stimulation is needed to selectively activate the somas [25]. Control of the electric stimulation parameters was proposed as a solution, which is discussed in the following section.

#### 3.3.2. Electric Stimulation Parameters for Selective Cell Activation and Chromatic Vision

An alternative approach to the selective activation of retinal cells is to manipulate the electric impulses sent by the electrodes [25]. That is, instead of manipulating the electric field shape and confinement to optimize selective cell activation, electrical stimulation strategies were proposed to optimally target specific cells and leave other cells inactivated. This stimulation strategy requires finding the pattern of electrical stimulation, including the amplitude and frequency of the electrical pulse, which permits targeted retinal activation. Then, these impulses are sent through the MEAs to activate the retinal cells. Two main reasons were suggested that favor this approach over electric field confinement. Firstly, researchers showed that there may be more than 40 types of RGCs, and each may be responsible for different aspects of visual information processing [50]. Selectively stimulating each of these types of RGCs would be desirable for the full restoration of vision [48]. Individualized electrical stimuli from the independent electrodes on MEAs can be used to target different types of cells, such as ON- and OFF- cells. Moreover, researchers found that the response to an electric pulse produced by RGCs is also dependent on the patient disease genotype [51]. Patients with varying genotypes of retinitis pigmentosa would, therefore, have varying quality of prosthetic vision, even though the same device may be implanted in all patients. Through their study, the authors asserted the importance of exploring novel stimulation strategies to enhance the response ratio between ON- and OFF- cell responses, such that these strategies can be customized for different genotypes. If only electric field confinement was used, it would not be possible to obtain the desirable output responses for the different cell types and different disease genotypes. The second reason for using electrical stimulation parameters in retinal prostheses is that researchers found that frequency-modulated electrical stimulation can provide hope for chromatic vision restoration in blind patients [52]. Stanga et al. (2011) first discovered that color perception could be evoked by changing the relevant stimulation parameters [53]. In their clinical trial, nine subjects with retinitis pigmentosa were equipped with the Argus II retinal prosthesis system [53]. The study methodology involved the simultaneous stimulation of various electrode pairs using different permutations of cathodic–anodic pulses [53]. The subjects reported eight different colors including orange, red, blue, green, and white; blue, yellow, and white were perceived the most [53]. A subsequent study in 2012 enrolled four blind subjects diagnosed with retinitis pigmentosa, who were equipped with the Argus II retinal prosthesis system [54]. The findings of this study demonstrated that it is possible to simultaneously evoke two distinct colored flashes, a significant difference from the first study [54]. In total, the patients perceived seven color combinations: gray–white, yellow–gray, orange–white, white–blue, brown–white, yellow–white, and yellow–blue [54]. The authors then concluded that manipulating the stimulation parameters can provide rudimentary color vision to blind patients and allow them to simultaneously perceive multiple colors, depending on stimulation parameters [53,54]. More recently, V L Towle et al. (2021) reached a similar conclusion and suggested that color hue development in prostheses may be dependent on stimulus intensity [55]. Paknahad and colleagues designed a study and were able to demonstrate an “amplitude-frequency” stimulation strategy to encode color vision, which was validated with experimental data [56]. Additionally, a case series study that followed seven subjects blinded by advanced RP and fitted with an epiretinal prosthesis showed that five/seven subjects perceived chromatic vision when frequency-modulated electrical stimulation of the retina was tested [52]. In all cases, color vision was achieved by selectively targeting small bistratified cells [52]. The knowledge that targeting specific cell types with specific parameters can restore some color percepts to patients creates an immense value for implementing the electric stimulation parameters that can selectively activate those cells. Moreover, unlike the electric field confinement method, the incorporation of electric stimulation can preferentially activate RGCs’ somas (versus passing axons in the nerve fiber layer) by manipulating the stimulus’ durations, phases, and waveforms [57].

While the two methods for selective stimulation are presented in comparison with each other, it should be noted that the use of one method does not exclude the use of the other. Indeed, both methods can and have been amalgamated to achieve higher selective activation. Activation through current steering avoids lateral stimulation and cross-talk, while using electric stimulation strategies can aid in targeting certain cells that are only responsive to that stimulus [30]. For instance, Jepson et al. (2014) investigated the usage of current steering in retinal prosthetics using multiple, densely packed suprachoroidal electrodes in the macaque retina, while providing simultaneous stimulation parameters for these electrodes [58]. Jepson and colleagues identified that when a current is applied to two adjacent electrodes, a peak current is produced at an equidistant point, termed a virtual electrode. Modulating the current ratio creates biased electric fields; in other words, if one electrode receives a higher current, the virtual electrode shifts toward that electrode. The authors predicted that by using current steering to optimize the stimulation intensity, frequency, and number of electrodes, it could be possible to increase target cell selectivity while reducing the response probability of non-target cells. Since it is challenging to document every stimulus pattern permutation, the authors proposed a piecewise linear model. They then measured the produced pattern and found that it was almost identical to the spatial pattern predicted by the model, “allowing for 0.90 [~90%] activation probability of the target cell with only 0.11 [~11%] probability of activating the neighboring cell [58]”.

### 3.4. Bidirectional/Closed-Loop Retinal Prostheses

One of the recent advances in the field of retinal prostheses is the design and development of bidirectional and closed-loop systems. These systems are able to both send electric impulses to retinal cells and record the electric responses evoked by stimulation [59]. The increased emphasis on this area in recent research can be attributed to several reasons. The first reason is the neural plasticity of the retina. With disease progression, photoreceptors within the retina can degrade. In response, the retina remodels, altering the electrophysiologic properties of the retinal pathways [60]. Due to this dynamic change in the electrophysiologic properties over the disease course and the consequent interpatient differences, the electrical patterns used to stimulate retinal cells need to be iteratively manipulated to produce an optimal outcome for patients [61]. For instance, as recently demonstrated by Caravaca-Rodriguez et al. (2022), the progressive degeneration of the retina resulted in an increased electrical threshold needed by subretinal and epiretinal prostheses [62]. The second reason driving innovation in the development of bidirectional systems is that these systems can enable fully representative, personalized, and iterative stimulation strategies. Often, to test stimulation parameters, including stimulus amplitude, width, and time, experiments are conducted in ex vivo conditions. These conditions, however, lack the metabolic environment of the eye, upon which the true electrophysiologic properties are dependent [59]. Thus, in vivo, bidirectional systems can provide a more complete reflection of the retinal response to stimulation and fill in any areas of uncertainty of ex vivo testing. Finally, bidirectional systems are important, since, as mentioned in previous sections, there is often a variable electrode–retina distance upon implantation [13]. This variability in electrode–retina distance impacts the stimulation ability and the produced phosphenes [45]. Thus, having closed-loop systems that guide the required stimulation of individual electrodes based on the evoked cell response can significantly improve patient outcomes. These closed-loop systems would enable such post-implantation customization. 

In addition to these factors, closed-loop systems also help address some of the previously mentioned biological challenges. For instance, Tandon et al. (2021) implemented bidirectional epiretinal prostheses to develop an algorithm for detecting axon bundle activation [42]. Similarly, Ghaffari et al. (2021) developed a closed-loop neural network (NN) model of RGC spatial activity and used it to create a real-time optimization method to search for stimulation parameters that elicit focal responses from in vitro retina [63]. 

The incorporation of bidirectional closed-loop systems into retinal prostheses requires an understanding of how the recorded cell response can then be used to feedback stimulation strategies to target cells [40]. As such, there has been an influx of computational models that allow researchers and engineers to understand the electrical activity of the retina and the impact that different stimulation parameters have on different cell types at varying levels of degeneration [64,65,66,67,68].

## 4. Clinical Considerations for Retinal Prostheses

### 4.1. Importance of Patient Selection and Screening 

The Argus II is currently the only FDA-approved retinal prosthesis device in North America. However, despite FDA approval, its effectiveness has not been confirmed. On the other hand, the Alpha AMS is CE-approved, and its efficacy is still being evaluated. This subsection specifically focuses on the Argus II and Alpha AMS.

Successful outcomes with retinal prostheses, such as the Argus II and Alpha AMS devices, rely heavily on proper patient selection and screening.

Indications for Argus II implantation include [69]

Aged 25 years or older;A prior history of useful vision;Profound visual loss resulting from the loss of photoreceptors (e.g., retinitis pigmentosa) that limits visual acuity to hand motion or bare light or no light perception in both eyes;A patient with no light perception must demonstrate retinal ability to respond to electrical stimulation, which can be confirmed through a dark-adapted flash test and visual evoked potential (VEP) testing;A patient must be in pseudophakic or aphakic status or have phakic status requiring cataract surgery or lensectomy prior to retinal prosthesis implantation;A patient must be able to attend post-implant clinical follow-up, device fitting, and visual rehabilitation.

Contraindications for Argus II implantation include [70]

Vision better than counting fingers in one of the eyes;Ocular conditions that prevent adequate visualization of internal structures, such as corneal opacification;Ocular conditions that affect device functionality, such as optic neuropathy, central retinal artery occlusion (CRAO), central retinal vein occlusion (CRVO), retinal detachment, severe strabismus, and amblyopia;Systemic conditions that contraindicate general anesthesia;Presence of metallic or other implantable devices in the head (e.g., cochlear implants);Hearing impairments that can interfere with a patient’s interaction with the Argus II device;Inability to comply with post-operative follow-up and rehabilitation due to cognitive decline or other conditions such as dementia or developmental disability.

The indications and contraindications of the retinal prosthesis Alpha AMS are similar to those of the Argus II, with the exception that there is no age specification mentioned in the literature for Alpha AMS implants. However, it is important to note that for the Alpha AMS, the retina must have a thickness of >100 μm to require functionality. More precisely, the indications for Alpha AMS implantation include [71]

Light perception without projection or no light perception in hereditary retinal diseases (retinitis pigmentosa, choroideremia, and Usher syndrome);Primary rod cone or cone rod dystrophies in their end-stage diseases;A prior history of normal visual function for >12 years.;A prior history of pseudophakia or aphakic status prior to retinal prosthesis implantation;A fluorescein angiography showing retinal vascular perfusion in all four quadrants of the macula;Evidence of inner retinal function (ganglion cells and optic nerve function) observed by the ability to elicit phosphene thresholds;Ability to give written informed consent and to attend follow-up and visual rehabilitation.

Contraindications for Alpha AMS implantation include [70,72]

Ophthalmic conditions with relevant effects upon visual function (glaucoma, diabetic neuropathy, retinal detachment, optic neuropathies, heavy clumped pigmentation at posterior lobe, and cystoid macular edema);Retina < 100 μm or no layering of the inner retina shown by OCT;Scar tissue (epiretinal, intraretinal, subretinal, and macular pucker);Occipital stroke;Congenital blindness and severe amblyopia;Substantial corneal opacity or any opacification of ocular structures that prevent clear image transmission;Active inflammation (uveitis);Systemic conditions that could pose significant risks during general anesthesia (cardiovascular/pulmonary/severe metabolic conditions such as diabetes);Life expectancy < 1 year;Inability to comply with post-operative follow-up and rehabilitation due to psychiatric/neurological diseases (Parkinson’s, dementia, MS, epilepsy, and severe depression and anxiety).

#### 4.1.1. Pre-Operative Assessment, Examination, and Imaging

A thorough pre-operative assessment is crucial for identifying suitable candidates for retinal prostheses. This assessment should include a clinical examination and ancillary testing. A clinical examination involves a complete ophthalmic examination, including an anterior segment examination and a dilated fundus examination. This examination helps to assess the overall ocular health of the patient and detect any contraindications to device placement. Optical coherence tomography (OCT), ultrasonography, and optical biometry for axial length measurements are also performed to help evaluate the retinal health and ocular anatomy, ensuring optimal conditions for device functionality [73,74].

#### 4.1.2. Post-Operative Rehabilitation

Post-operative rehabilitation plays a significant role in maximizing the benefits of retinal prostheses. Patients must learn to interpret the artificial vision generated by the device, which is different from natural sight. This situation is comprehensively covered in the following sections of this article. Rehabilitation focuses on enhancing the patient’s quality of life and independence by teaching them to use the limited vision provided by the device in conjunction with their existing auditory and tactile function skills. Family members and a social support network are crucial in facilitating the rehabilitation process, by providing encouragement and motivation to the patient [75,76].

As with any surgical procedure, it is valuable to weigh the benefits of treatment in comparison to the possible risks. The following section aims to provide a risk-to-benefit analysis for retinal prostheses by discussing their clinical outcomes.

### 4.2. Safety and Adverse Events

#### 4.2.1. Epiretinal Prostheses

Shaffrath et al. (2019) conducted a study on the Argus II prosthesis system, in which 47 adults were followed for a period of 12 months. The study revealed a total of 13 serious adverse events, of which 9 were related to the implant, and 4 were related to the procedure. The reported adverse events were conjunctival erosions (n = 4), hypotony (n = 2), and explantation (n = 2) [77]. Other adverse events included ocular inflammation (n = 1) as well as tractional (n = 1) and rhegmatogenous retinal detachment (n = 1) [78]. While, in total, 13 adverse events were observed, 11 of these were resolved while 2 cases persisted (hypotony with a duration of eight months and the rhegmatogenous retinal detachment, which was deemed permanent) [78]. In another study by Stanga et al., which studied the same implants, the authors followed five patients who received implants. Out of those, three cases of retinal detachment were reported; however, all of them were successfully treated [77]. Finally, in a more recent (2021) study by Delyfer et al., which followed 17 subjects over two years after implantation of the Argus II, the authors reported two serious adverse events. One of these included a vitreous hemorrhage, a consequence of the device itself, and the other was endophthalmitis, a consequence of the implantation procedure. Other adverse events included macular edema (n = 1), choroidal detachment (n = 1), conjunctival cyst (n = 1), keratitis (n = 1), bilateral phlebitis (n = 1), and ptosis (n = 1). All the adverse events were resolved, except for the ptosis [79].

The IRIS 2, another epiretinal device, was the subject of clinical trials that reported a total of 17 adverse events. Of these, 11 were non-serious adverse events that included phlebitis (n = 1) and tack detachment requiring refixation (n = 1). All the non-serious events were resolved. The remaining six events were classified as serious and included hypotony, secondary to fluid leak from the sclerotomy site (n = 2); vitreoretinal preretinal traction (n = 1); and persistent pain (n = 1) [80].

Epiretinal device implantation is a procedure that is similar to routine vitreoretinal surgery. Due to this similarity, the process of implantation and the explanation of the procedure are relatively simple. As a result, it is not surprising that the overall rate of complications associated with this type of surgery is well-controlled [81].

#### 4.2.2. Subretinal Prostheses

Subretinal implants were reported to be technically more challenging due to the presence of adhesions between the retina and retinal pigment epithelium from degeneration and lack of familiarity of the surgical approach [82]. 

Older studies on Alpha IMS subretinal implants noted adverse events that included increased intraocular pressure resolved without sequelae and retinal detachment resolved with explantation, with retinal fibrotic changes [83]. A study evaluating the safety of a newer edition, the Alpha AMS, reported four cases of surgical dehiscence, two cases of implant displacement, one case of partial loss of the silicone oil tamponade (n = 1), and one case of pain (n = 1) [72]. Once again, all these complications were self-limiting or surgically well-managed. 

A study evaluating the PRIMA device, a subretinal retinal prosthesis, reported adverse events related to procedural complications. These events included a choroidal hemorrhage, secondary to inadvertent insertion into the choroid, caused by patient movement under local anesthesia; focal subretinal hemorrhage; and the displacement of the device following non-adherence to post-op movement limitations. Post-op medication non-adherence also led to increased intraocular pressure in one patient that was successfully treated [28]. Adverse events of another subretinal device, the Alpha AMS, included two cases of conjunctival erosion, one case of peripheral retinal detachment, and one case of contact dermatitis, which were subsequently well-managed [84].

#### 4.2.3. Suprachoroidal Prostheses

Suprachoroidal implants are unique in that they are considered less technically and surgically challenging compared to other implant types, and they limit the need for retinal incisions [85]. Recent clinical trials following patients 56 weeks post-implantation found expected non-serious adverse events such as pain, swelling, tenderness, conjunctival injection, ocular pressure sensation, and local inflammation [86]. One case of increased ocular pressure associated with topical steroid use for eyelid edema was reported but was resolved by weaning of the medication and appropriate treatment [86]. These findings of limited complications appeared to echo an earlier study on suprachoroidal devices [85]. 

All in all, these overall trends corresponded with older reports of adverse events in retinal prostheses, where events were either few or well-managed [87]. The complication type and frequency corresponded to the categories of implants according to the invasiveness of the procedure, with suprachoroidal retinal prostheses reporting the least adverse outcomes. Nonetheless, retinal prostheses continue to be safe and effective tools for restoring visual function in individuals with macular degeneration.

### 4.3. Visual Function and Outcomes

#### 4.3.1. Epiretinal Prostheses

Recent studies demonstrated interesting findings regarding visual function. A study on the Argus II analyzed visual function through localization squares generated at random and direction of motion. While some demonstrated improved visual function, an important proportion saw no difference or benefit from having the device off [78]. More specifically, square localization found 45% benefitting versus 55% having no difference or benefit with the device off, while the identifying the direction of motion results were split, 35% versus 65% [78]. One way that researchers evaluate the functional outcomes of the device is through the use of a self-report questionnaire, called the functional low-vision observer-rated assessment (FLORA). The researchers that studied epiretinal retinal prosthetic use in individuals with age-related macular degeneration found similar mixed benefits on screen-based tasks but noted improvements on all tasks evaluated by the FLORA with the device on versus off [77]. On the other hand, the evaluation of the visual function in the IRIS 2 saw an improvement in the square localization and direction of motion detection at the 3-month and 6-month marks. This same study also found enlarged visual fields and enhanced picture recognition with the device on [80].

#### 4.3.2. Subretinal Prostheses

The investigation of basic visual screen tasks with the Alpha IMS displayed improved scores in light source perception in 86% of participants, which dropped to 59% when it came to localizing, while motion detection was more difficult, with only 21% passing, including one that attributed their success to chance [83]. The facilitation of daily living activities such as identifying items on a dining table, telling the time, and reading letters with the device on were also evaluated in this study. In total, 45% found the device to be useful, 27% found little benefit, and 28% saw no benefit in completing these tasks [83].

Edwards et al. investigated the Alpha AMS and found improvement in tabletop object identification and grayscale contrast detection; however, they found that participants continued to have difficulty telling the time with or without the device. As for screen-based tasks, participants of this study found more success with light localization and perception with the device on; whereas for motion detection, there was no benefit from having the device on versus the device off [84]. Analysis of the PRIMA subretinal device also found improvements in eccentric natural acuity and the accurate identification of bar orientation [28].

#### 4.3.3. Suprachorodial Prostheses

A study investigated the improvement in task accomplishment over two years in patients with late-stage retinitis pigmentosa and found that the extent to which suprachoroidal retinal implants facilitated daily activity varied. Tasks relying on vision, such as washing dishes, folding and organizing laundry, and identifying doorways and people in non-crowded spaces showed the greatest benefits over time, with peak improvement between 17 and 44 weeks post-implantation [88]. Of note, identifying food on a plate remained very difficult, attesting to the device’s limited visual information and quality [88]. On the other hand, walking around the home and independently walking up and down stairs were reported as having no difference with or without the use of the device [88]. Laboratory-based functional visual tasks were also evaluated. Of note, participants scored significantly higher on square localization and displayed improved motion discrimination with the device on versus the device off [86].

Individuals with prosthetic vision also demonstrated oculomotor behavior post-implantation in motion discrimination tasks with smooth pursuit and an altered opto-kinetic reflex, which presented with upbeat nystagmus regardless of the stimulus motion [89]. The presence of these “naturalistic oculomotor responses” left room to further develop these devices by augmenting their complexity and incorporating this finding [89]. 

In summary, studies on retinal prostheses, including epiretinal, subretinal, and suprachoroidal devices, showed mixed results in improving visual function and facilitating daily activities in individuals with macular degeneration. While some individuals may see improvements in tasks such as square localization and the direction of motion detection, others may not see a significant difference in visual function with the device off. Self-report questionnaires such as the FLORA showed improvements in tasks evaluated with the device on versus the device off, but the benefits may vary depending on the specific functional tasks being evaluated and the type of prosthesis being used. Further research is needed to fully evaluate the potential benefits and limitations of retinal prostheses and to optimize their use in clinical practice. Additionally, studies showed that retinal implants can generally facilitate daily activities such as washing dishes and identifying doorways and people in non-crowded spaces, while identifying food on a plate remains difficult.

Table 1 summarizes the adverse events, visual function, and outcomes of the different retinal prostheses available to patients.

### 4.4. Rehabilitative Programs

As part of the clinical implementation of retinal prostheses, rehabilitative programs are essential to improve patient outcomes. Retinal prostheses showed a steep learning curve, requiring resiliency from patients to achieve success [41]. Users of retinal prostheses often described the visualized phosphenes as stressful and cognitively fatiguing [41]. Without rehabilitation, recipients sometimes stop using the device within years, due to reasons such as user familiarity with their home environment and, thus, no longer needing the device; general disappointment; or the inconvenience of the device [41,90]. Furthermore, even with adequate disclosure about the difference from natural vision, participants remained unprepared for how different this vision was from what they had once known [41]. Those that reported continued use of the device adjusted the conditions to decrease the intensity and activity of the signals or used the device as visual entertainment [41].

Additionally, there are often obstacles that interfere with vision and emphasize the importance of rehabilitation. These include a phenomenon where phosphenes persist for a duration after a stimulation is complete or another phenomenon where they fade in certain areas of the retina even with stimulation, possibly due to desensitization of the RGCs that requires correction with head movements. Phosphene persistence and perceptual fading onset differ from person to person, so it is suggested that these factors should be taken into account during patient rehabilitation in order to optimize prosthesis’ efficacy [91]. While some modifications to the technology were proposed, particularly with regard to specific spatiotemporal stimulation strategies, as mentioned earlier, the strategies implemented throughout rehabilitation training are considered valuable to counteract some of the current challenges posed by prostheses [92]. It was found, for instance, that retinal prostheses users were able to improve visual search and perception with auditory cueing by emphasizing the importance of including audio-visual training in future rehabilitation protocols [93]. Additionally, postoperatively, the use of head movements was recommended in certain prostheses to “refresh the visual field viewed by the camera”, allowing for improved outcomes [92].

A pilot study looked at the feasibility of the Computer Assisted Rehabilitation Environment System (CAREN), an interdisciplinary approach comprised of visual exercises and dual-task training (i.e., motor and visual tasks) adjusted to individual ability, with the goals of improving visual function and safety [94]. Following biweekly 60-min sessions for four weeks, the pilot study found CAREN to be safe, feasible, and effective in improving functional outcomes [94]. The importance of a multidisciplinary approach in order to manage patient expectations was also echoed in other case studies [90].

## 5. Comparison of Alternative Emerging Therapies to Retinal Prostheses

Retinal prostheses have emerged as a promising technology for restoring vision in patients with retinal diseases. However, the safety and efficacy of these devices are still being evaluated through clinical trials, and their high cost limits their widespread adoption. Considering these limitations, it is essential to explore the alternative therapeutic approaches on the horizon for treating retinal diseases. In this section, we provide a brief overview of some of the latest developments in cell-based and gene-based therapies for retinal diseases. By examining these new treatments, we aim to assess whether retinal prostheses are losing favor as a therapeutic option.

### 5.1. Cell-Based Therapies

Cell-based therapies typically involve the use of stem cells to replace or restore dysfunctional cells in the retina, often with the help of trophic factors to promote cell survival and growth [95]. Several types of stem cells have been explored as potential sources for cell-based therapies, including pluripotent and embryonic stem cells, bone marrow stem cells, stem-cell-derived retinal progenitor cells, and retinal pigment epithelium (RPE) cells. 

Pluripotent stem cells, which are capable of differentiating into any cell type in the body, offer a promising option for cell-based therapies in retinal diseases. While human embryonic stem cells (hESCs) have been a commonly studied type of pluripotent stem cell, their use raises ethical concerns, as they are derived from human embryos. Alternatively, induced pluripotent stem cells (iPSCs) can be generated from somatic cells, avoiding the ethical concerns associated with hESCs. iPSCs can be directed to differentiate into retinal pigment epithelium (RPE) cells, which play a crucial role in supporting photoreceptor function in the retina. In retinal diseases where the RPE layer is damaged, the replacement of this layer using PSC-derived RPE cells showed potential to delay disease progression and restore vision loss [96]. 

Bone marrow stem cells (BMSCs), including mesenchymal stem cells (MSCs) and hematopoietic stem cells (HSCs), are extracted from the bone marrow. Since BMSCs are multipotent stem cells, their capacity of differentiation is more limited than that of PSCs. However, BMSCs are able to generate cells that are specific to a certain type of tissue. Setting BMSCs apart is their ability to transdifferentiate, to migrate toward lesion sites [97]. It has not yet been proven that BMSCs can fully differentiate into photoreceptors, but recent preclinical studies showed their ability to release anti-angiogenic and neurotrophic factors and many other molecules such as insulin-like growth factor-1 (IGF-1), Th2-related cytokines, and the class II major histocompatibility complex (MHC II) [98]. 

As previously mentioned, damage to the RPE layer affects photoreceptor function and can lead to vision loss. Stem-cell-derived RPE offers an interesting treatment option by providing trophic support to the remaining photoreceptors [99]. Since RPE is composed of a single layer with a morphology and a physiology that are well-known, it becomes a relatively good candidate for stem cell therapies. Furthermore, because of its small size and given its immune privilege, the eye offers a great microenvironment for stem cell therapies [100]. By simultaneously transplanting fetal retinal pigment epithelium and MSCs in mice, their electroretinograms (ERGs) were found to have improved. Due to the expression of a protein-activating rhodopsin (Crx), an increase in rhodopsin levels, and a decrease in caspase-3 expression, the survival rate of the transplanted cells was shown to significantly improve. Therefore, coculture transplantation proved to be better than single-cell transplantation, which opens up future possibilities for the development of better treatments for retinal diseases [101]. Stem-cell-based therapies show potential as a treatment option for retinal diseases caused by RPE dysfunction and loss, such as age-related macular degeneration (AMD) and Stargardt’s macular dystrophy (SMD). While the safety of stem-cell-derived RPE transplantation has been confirmed, the efficacy in terms of visual outcomes still requires further investigation. The timing of RPE transplantation is crucial for the success of the procedure, as it plays a key role in preserving photoreceptor function. Researchers suggest that RPE transplantation should occur following RPE dysfunction but preceding photoreceptor injury, in order to achieve optimal visual outcomes. However, stem-cell-based therapies require systemic immunosuppression to prevent rejection, which may limit their use in patients with other comorbidities. As a result, many patients may not be candidates for this therapy [100]. 

The developing neural retina of human fetuses aged between 16 and 20 weeks of gestation generates another type of stem cells: retinal progenitor cells (RPCs) [102]. These mitotically active MSCs were shown to express photoreceptor markers in vitro, which would enable them to differentiate into the neural cells of the retina [103]. RPCs stand out by their ability to secrete trophic factors that would increase retinal survival and enable researchers to replace photoreceptors [104]. With ongoing preclinical studies, stem cells have proved themselves as having the potential to regenerate the retina. Given their ability to turn RPCs into photoreceptors and to form functioning synapses with the host, stem cells could possibly stabilize or even reverse vision loss. 

An example of how stem-cell-based implants have contributed to vision restoration is the “California Project to Cure Blindness Retinal Pigment Epithelium” (CPCB-PRE1) [105,106]. This implant system is composed of a monolayer of human embryonic stem cells, which are derived from retinal pigment epithelium cultured on a media that mimics the Bruch’s membrane. CPCB-RPE1 implants were shown to improve vision and slow down progressive vision loss in a phase I/IIa clinical study [107].

### 5.2. Gene-Based Therapies

Gene therapy has emerged as a promising approach for treating retinal diseases such as retinitis pigmentosa (RP) that are caused by genetic mutations. By targeting the root cause of the disease, gene therapy offers the potential to restore vision in affected individuals. Different approaches can be used depending on whether the inherited retinal disease is caused by a recessive or dominant genetic mutation. 

For recessive mutations, a gene complementation approach is typically used, while for dominant mutations, gene suppression with or without gene complementation is generally preferred. To implement gene therapy, various techniques were developed, including the use of viral and non-viral vectors and CRISPR-cas9 gene editing. These approaches allowed for targeted delivery and the expression of corrective genes to the retina [108]. 

Viral vectors used for gene therapy can be divided into three subtypes: adenovirus (Ad), adeno-associated virus (AAV), and lentivirus (LV). Of these, AAV attracted the most attention for treating inherited retinal diseases due to its small size (25 nm), which allows for efficient targeting of the retinal layers, and low immunogenicity, which makes it a good candidate for long-acting treatment [109]. However, AAV has limited capacity to clone a large genome (packaging capacity of 4.7 kb), which restricts its use to treating recessive diseases such as Stargardt’s disease [109]. 

Non-viral vectors were also studied for gene delivery, as they reduce the risk of an immune response and can deliver larger genes compared to viral vectors. However, non-viral vectors have not yet been shown to efficiently deliver genetic material into cells, unlike viral vectors that have developed efficient ways for nucleus entry [110]. 

The gene-editing system known as CRISPR-cas9 has emerged as an important tool for gene expression and suppression. It showed promising results in silencing dominant mutations using the non-homologous end-joining pathway in inherited retinal diseases. However, the success of CRISPR-cas9 is highly dependent on guide RNA design, the type of target cell, whether a viral or non-viral vector is used, and host-related factors [111].

Currently, the only gene therapy approved for treating a retinal disease is Luxturna, which targets the RPE65 gene and uses an AAV vector to treat Leber Congenital Amaurosis type 2 (LCA2). LCA2 is a good candidate for gene therapy, as it involves an enzyme expressed in the RPE layer, which is an easier target for AAV than the photoreceptor layer. However, most other IRDs involve gene transfer to the photoreceptor layer, which poses a challenge for gene therapy [110]. 

Despite the potential of gene therapy for treating IRDs, its application in practice is complicated by the wide variety of gene mutations causing these diseases. An in-depth understanding of pathogenesis is required, and further research is still needed to assess the long-term outcomes and safety of the therapy [112].

### 5.3. Optogenetics

Another field that recently gained attention for treating inherited retinal diseases (IRDs) is optogenetics. Unlike gene therapy, which aims to restore the function of mutated genes, optogenetics involves introducing photosensitive optic proteins (i.e., opsins) to the degenerated retina to restore function and provide photosensitivity to the remaining retinal cells. The primary goal is to turn non-light-sensitive cells, such as bipolar and retinal ganglion cells, into photoreceptors. 

Two types of opsins are used: microbial opsins (type 1) and animal opsins (type 2). The choice of opsin depends on the desired effect, with depolarized opsins mimicking an “off” response in dormant cells and hyperpolarized opsins mimicking an “on” response [113]. The two types of opsins differ in light sensitivity, function, and utility for vision restoration [114]. Type 1 opsins cause a conformational shift and directly affect ion channels or pumps, while type 2 opsins indirectly affect ion channels through intracellular G-protein-coupled receptor (GPCR) signaling cascades when absorbing light [115].

Currently, there are phase I/II clinical trials underway for optogenetic therapy, since proof of concept was established. However, further research is needed to understand the structure, molecular transport modes, dynamics, and optical properties of photosensitive proteins [112].

### 5.4. Verdict

After reviewing the different treatment options for inherited retinal diseases (IRDs), it is possible to consider whether retinal prostheses may become less attractive compared to cell-based and gene-based therapies. However, each therapeutic avenue has its strengths and challenges, and having more options available means that each option’s strengths can be used to overcome others’ weaknesses. For example, a study by Nascimento-dos-Santos et al. demonstrated the synergy between gene therapy (AAV2-pigment epithelium-derived factor (PEDF)) and cell therapy (human mesenchymal stem cell) in increasing Tuj1-positive cells, demonstrating retinal ganglion cell (RGC) layer neuroprotection and promoting axonal outgrowth. Combining these therapies provided greater neuroprotection and axonal outgrowth compared to gene therapy alone [116]. 

Optogenetics may also be an interesting solution for improving retinal prostheses, as noted in a study by Lagali et al. Optogenetic prostheses would not require invasive surgery, and the need for biocompatibility would be reduced. Using optogenetics could also be a possible solution for overcoming the challenge of the low restoration of vision seen in retinal implants, since optogenetics would only require a high brightness display and the right optics [117]. Additionally, a study by Gongxin Li et al. showed that ChR2-expressing cells could produce a mechanically responsive signal out of a light stimulation by using the size of cellular deformation as an indicator. This study demonstrated that implants integrating optogenetics offered better resolution than existing implants, potentially opening up the possibility of a new generation of retinal prostheses [118]. 

While retinal prostheses face challenges in terms of vision quality and surgical risks, combining them with other treatment options such as cell, gene, or optogenetics therapy could potentially provide greater neuroprotection and improve the quality of vision restoration. Therefore, the continued research and development of various therapeutic avenues is essential to provide the best possible treatment for inherited retinal diseases. To achieve this, interdisciplinary collaboration is essential between ophthalmologists, geneticists, stem cell biologists, and engineers, among others, to continue developing and refining therapeutic approaches that can provide optimal patient outcomes.

Table 2 provides a comparative overview of the emerging treatment modalities for retinal diseases, including retinal prostheses, cell-based therapies, gene-based therapies, and optogenetics, highlighting their core mechanisms, potential advantages, and current limitations.

## 6. Recent Advancements in Retinal Prosthesis Technology

### 6.1. Advances in Engineering of Prostheses

Retinal prostheses must meet challenging design criteria. Researchers have pioneered advancements in the field, such as improvements in material science, visual field size, and the integration of artificial intelligence for optimizing image processing. Some of the prominent advances are discussed in this section to reflect on the progress and innovation in the field, leading to a discussion on the outlook of retinal prostheses.

#### 6.1.1. Material Science

There are two categories of electrode materials proposed for retinal prostheses: inorganics—such as metals, silicones, and carbon-based materials, and organics—such as polyimide and polydimethylsiloxane (PDMS) [119]. The literature consistently demonstrated that smaller-sized electrodes optimize electrical signaling due to comparable dimensionality to their neural targets; however, the innate physical limitations with current electrodes require further inquiry [120]. 

Metal microwires typically involve conducting metals; in fact, the five most commonly used metals in retinal prosthetics are iridium, gold, titanium, tin, and platinum [121]. They are chemically inert, have desirable electrical properties, and have tested biocompatibility and applicability in neural interfaces [119]. Planar microelectrodes are commonly used for retinal prosthetics due to their relatively high spatial resolution capacity [119]. However, metal microwiring has certain intrinsic limitations, namely, its propensity for high impedance and lower charge storage capacities [119]. Moreover, its rigidity makes it challenging to fixate, and the need to assemble transcutaneous wire connections increases the probability of adverse surgical and long-term progressive physiological complications [119]. 

Microelectromechanical system (MEMS)-based silicon materials may be considered to combat the above issues. These techniques adopt microfabrication techniques that can conform a rigid planar structure into context-specific layouts. Consequently, they allow for higher density recording sites [122]. However, a significant drawback with these materials is a lack of conformity between the rigidity of the silicon electrodes and native tissue. Furthermore, even after reducing the electrode thickness to improve bending rigidity, the smallest MEMS-based electrodes are still too large for optimal conformability [119]. For the above reasons, conductive polymers, carbon-based materials, and nanomaterials are promising alternatives. 

Conductive and semi-conductive polymers are organic materials with intrinsic electrical and optical characteristics. They demonstrated efficacy in electrode–tissue interfaces and were better than purely metal-based materials because of their chemical stability, improved biocompatibility, and high conductivity [123,124]. The most popular examples include poly(3,4-ethylene dioxythiophene) (PEDOT), polyaniline, polythiophene, and polypyrrole (PPy)—the lattermost being the most used in brain–machine interfaces [119]. Similarly, PEDOT treated with polystyrene sulfonate also demonstrated superiority on the previously mentioned metrics [124]. However, due to the weak interactions of the conductive polymer coating with the electrode, there was a high risk of delamination—areas where there is little to no bonding between the coating and the electrode—causing disruptions in how the signal is registered [125].

Ouyang and colleagues (2017) proposed surface functionalization–modifying the surface properties of the electrode–as a solution to these issues [126]. In their investigation, they functionalized PEDOT with (2,3-dihydrothieno [3,4-b] [1,4] dioxin-2-yl) methenamine (EDOT-NH2)—a methylamine-functionalized EDOT derivative—onto a metal substrate using electrografting techniques. Electrografting involves creating strong covalent bonds between organic materials with solid substrates. Their findings demonstrated better adhesion with the P(EDOT-NH2) anchoring layer and improved electrode durability.

Green et al. (2010) demonstrated that PEDOT was significantly more efficient at charge transfers compared to traditional metal electrodes [127]. The first aspect of their study was in vitro, coating PEDOT on platinum microelectrode arrays and performing biphasic stimulation protocols. Their preliminary findings demonstrated a 15-fold greater charge injection limit than platinum, and PEDOT significantly reduced the potential excursion at the platinum electrode. Upon completion of the investigation, PEDOT also demonstrated a reduction in possible excursions when implanted into the suprachoroidal space of a cat retina.

Carbon-based materials are of growing interest in the literature because of their low modulus and electrical impedance [119]. Carbon nanotubes (CNT) accommodate sizeable effective surface areas in electrode–tissue interfaces; as a result, they provide greater charge transfer capacity and low interfacial impedance [128]. Additionally, Eleftheriou et al. (2017) assessed the structural and functional integration of CNTs in retinal prosthetics [129]. Over three days, stimulation thresholds decreased, and cellular recruitment increased [129]. These results indicated an increase in CNT–electrode–RGC coupling, while bypassing a negative glial response [129]. These novel findings indicate a potential therapeutic benefit of using CNTs in retinal prosthetics.

A lot is still unknown about the application of CNTs in retinal prosthetics; the most prominent criticism against them and other carbon-based materials, such as graphenes, is their potential for biotoxicity [119]. However, like conductive polymers, CNTs and graphenes can be functionalized to make them more biocompatible and optimize electrode–tissue interfaces [130,131]. In fact, graphene-based materials show promise as a viable alternative to current brain–machine interface electrodes because they are conformable, can be easily functionalized, are mechanically stable, and offer excellent electrical conductivity [131,132]. Moreover, Nguyen et al. (2021) performed an in vivo biocompatibility study using a novel graphene electrode on P23H rat model retinas [37]. Their results demonstrated a reduction in inflammation—as defined by microglial labeling—in the graphene-based electrodes compared to the biocompatible polymer-based electrodes [37]. However, further investigations assessing graphene-based electrodes on stimulation capacities in retinal prosthetics are needed to evaluate efficacy.

Nanowires were another pillar of investigation, given their ability to stimulate neurons at the axonal and dendritic levels [119]. Studies investigating the applicability of two- and three-dimensional nanowire electrodes demonstrated improved sensitivity and signal-to-noise ratio by virtue of an increased surface-to-volume ratio and increased density of neuron–nanowire electrodes [133]. Yang and colleagues (2019) discussed a novel biomimetic electrode technology, called neuron-like electronics (NeuE), which are probes that have high structural and mechanical comparability to neuron targets—the more significant variance between the neuron and electrode occasions disruptions in signal recording and damage to surrounding tissue [134]. Preliminary studies of NeuE in cerebral mouse models also demonstrated minimal immune provocation [134]. In addition, NeuE also showed some pro-regenerative properties; like a substrate, it seemed to be able to direct cells toward damaged tissue sites in need of repair [134].

#### 6.1.2. Visual Field Size

Advancements have recently been made in improving the visual field (or angle) size of retinal prostheses. Felauto and colleagues in Switzerland developed a novel foldable and photovoltaic wide-field epiretinal prosthetic called POLYRETINA [22]. POLYRETINA significantly improved retinal prosthetic technology because it improves the currently limited visual field [22,135]. A wide visual angle is required for optimal visual perception; it is involved in a myriad of processes ranging from attention and spatial recognition to estimating the layout space and interacting with the environment [136]. Therefore, for retinal prosthetics to demonstrate better artificial vision, they must allow the recipient to have a large visual angle size (VAS) [136].

Visual angle directly correlates with retinal coverage; however, current prosthetics, such as the Argus II, provide a VAS of up to only 20 degrees [23,137]. This coverage is inadequate for mobility tasks and navigation-related activities such as object identification [137]. Hence, current prosthetics require patients to scan their environment to continually mitigate the restrictions in VAS—the scanning and continual processing of visual information are known as space decomposition [137]. Patients report that the device’s need for continual space decomposition is mechanically and cognitively cumbersome [137]. Preliminary studies investigating the minimal visual field requirement to perform daily tasks—using simulated prosthetic models in sighted individuals—is approximately 30 degrees, but this is an underestimate [137]. Due to individual variations in vision and learning, a higher VAS is required to appropriately adapt to the prosthetic [137]. Current retinal prosthetics are lacking because they are too small, stimulate a small portion of the retina, and are difficult to surgically fixate on the retina [137].

In addition to improving VAS, POLYRETINA improves the spatial resolution of the prosthetic, as its wireless photovoltaic technology accommodates higher electrode number, density, and coverage to optimize artificial vision [23]. The retina’s function is to absorb the incident light entering the pupil and transmit that information to the brain as electrical and chemical messages to inform visual image processing [136]. POLYRETINA’s wireless photovoltaic technology serves a similar function: artificial light is projected onto the retina and absorbed by a semiconductor layer embedded into the stimulating electrodes [22,136]. Each pixel in the device converts incident light into information that is later processed by the brain [22,136]. The device contains 10,498 photovoltaic pixels and is densely packaged over an approximate area of 43 degrees [22,136]. Increased electrode density reduces the need for frequent environmental scanning and significantly improves spatial resolution [23]. Current prosthetics typically directly activate the nerve fiber, but this can alter the retinotopic map due to the residual activation of surrounding axons [137]. However, with POLYRETINA, direct activation is bypassed through network-mediated stimulation, preventing visual distortions [137]. In addition, the prosthetic can accommodate a high electrode number and density because its wireless capabilities omit the need for space-consuming hardware such as trans-scleral connectors, pulse generators, and other cables within the array [137]. While a high pixel density does not necessarily correlate with higher visual discrimination, emerging studies using POLYRETINA in ex vivo mouse models with retinitis pigmentosa demonstrate its ability to produce a high spatial resolution [22].

Another advantage of using a conformable device such as POLYRETINA is the ease of surgical fixation [23,137]. Implants designed to provide a greater VAS require large implants, producing more invasive incision sites. POLYRETINA, on the other hand, is foldable and injectable, and it only requires a 6 mm scleral incision. The added benefit of high conformability makes the device a safer option than non-conformable implants. In addition, wired implants observe greater risks of lead-wire damage—leading to implant dysfunction—and tissue scarring due to the added pressure of the mechanical wiring. POLYRETINA’s wireless technology bypasses these issues by eliminating the need for superfluous hardware.

#### 6.1.3. Artificial Intelligence

Artificial-intelligence-based imaging protocols are emerging as promising solutions to combat low sampling resolution. While phosphene optimization is an ongoing area of inquiry, advancements in AI research demonstrate the potential for devices to adopt computer vision algorithms to conduct preprocessing maximization of image quality. For instance, Ge et al. (2017) developed an obstacle-avoidance AI system, called NeuCube, using a spiking neural network [138]. The NeuCube architecture conducts real-time obstacle analysis and provides meaningful guidance to prosthetic wearers. In addition, it is significantly more stable and accurate than current navigation systems and does not rely on sensors or additional image-capturing hardware.

Deep learning models are now becoming integral to improving saliency mapping. DeepGaze II, created by Kümmerer et al. (2016), is a trailblazer in this space [139]. This model leverages developments in deep neural networks—namely VGG-19, a trained convolutional neural network (CNN)—to recognize objects and improve saliency prediction [139]. One area of improvement with deep learning models is integrating depth estimation protocols [140]. However, collecting per-pixel ground truth depth information to inform supervised learning is cumbersome [140]. Godard and colleagues (2019) developed monodepth2, a self-supervised model that can approximate per-pixel monocular depth, to address this gap [140]. It demonstrated significant improvements in reducing the presence of visual artifacts, reducing pixelations that infringe on camera motion assumptions, and reducing reprojection loss [140]. The caveat with the above-mentioned models is the scarcity of their usage in validated computational models of the retina. Han et al. (2021) used different deep-learning-based scene simplification algorithms on psychophysically validated retina models to address this deficiency [141]. Their findings revealed that these models could be utilized to accurately predict stimulation patterns, which would help inform refined algorithms to evoke responses at the desired tissue targets [141].

While CNNs, such as DeepGaze II, provide advanced improvements in two-dimensional image processing, they are inefficacious when consolidating three-dimensional videos [142]. An emerging alternative is a convolution recurrent neural network (CRNN), which integrates convolutional and recurrent processing to optimize spatiotemporal feature extraction. However, the recording and processing framework utilized by CRNN is energy-demanding, making it unfeasible for application in retinal prosthetics. Wang and colleagues (2022) developed an energy-efficient version called SpikeSEE, which uses a dynamic scene-processing framework [142]. It has improved prediction accuracy compared to a traditional CRNN, and it uses a spike processing protocol that helps decrease power consumption.

#### 6.1.4. Preserving Residual Visual Field

Currently, one of the primary concerns when implanting retinal prostheses is the possibility that they would impact the residual, functional, retinal cells in patients who have remaining peripheral vision. This concern is significant, in that it directly limits the scope of retinal prostheses, since those who have not had complete vision loss would be ineligible or unenthusiastic for implantation. Recently, a study was completed to improve central vision in atrophic age-related macular degeneration without threatening the residual peripheral field [28]. This study was carried out in humans and aimed to develop a wireless photovoltaic retinal implant (PRIMA) where pixels can convert images that are projected from video glasses using near-infrared light into electric current to induce inner retinal neurons without disrupting peripheral vision [28]. The clinical trial was performed on five patients suffering from a geographic atrophy zone of at least 3 optic disc diameters with no foveal light perception. By using a subretinal, 2mm wide, 30 μm thick chip that displayed 378 pixels implanted in the area of atrophy, it was shown that, in all five patients, the prosthesis was successfully implanted under the macula, and the photovoltaic subretinal implants were able to demonstrate visual acuity up to 20/460. All five patients were able to perceive white–yellow prosthetic visual patterns with adjustable brightness. The study also demonstrated that the implantation of PRIMA did not decrease the residual natural acuity in any patient [143]. Since the PRIMA did not negatively impact vision, further studies were conducted to improve PRIMA durability and estimate the device’s lifespan in vivo. These studies showed that these implants are robust, with in vitro reliability of at least 10 years, and are resistant to corrosion and water ingress [144]. Another clinical trial involving retinal implants was based on investigating the changes in neurosensory macular structures and the thickness associated with subretinal implantation in geographic atrophy [145]. In this study, changes in distance between electrodes and the retinal inner nuclear layer (INL) and alterations in the thickness of retinal layers were assessed using optical coherence tomography near the implanted subretinal chip within the atrophic area. It was shown that, in three out of five patients, the distance between the implant and the target cells remained stable over a long-term follow-up of 36 months [145]. The total retinal thickness located above the subretinal implant decreased on average by 39 ± 12 µm for 3 months post-implantation. However, it was also reported that no significant changes were observed after that. In addition, the retinal thickness decreased near the temporal entry point areas located outside of the subretinal implantation, following the surgical trauma of retinal detachment. All in all, this study demonstrated that the surgical delivery of photovoltaic subretinal implants promoted minor retinal thickness three months post-implantation, which remained stable over the long term with no disadvantageous structural or functional effects.

### 6.2. Outlook on Retinal Prostheses

Retinal prosthesis technology has been gaining much interest in recent years and is continuously for the target of further improvements [146]. As mentioned in earlier sections, the engineering design criteria need to be finely balanced to achieve the most optimal patient outcomes. A retinal prothesis can be influenced by several factors such as the electrode–retina topographical alignment, electrode size and material, factors that impact spatial selectivity, and, finally, incorporation of bidirectional systems. These factors should be considered when improving the previously reported, low visual acuities [147]. All these setbacks demand more research in the field of retinal implants, with a particular focus on visual processing in the retina and psychophysical performance.

The clinical application of retinal prostheses faces several bottlenecks that limit their widespread use. These include the following:–Limited effectiveness: While retinal prostheses can provide some degree of visual perception, the quality and resolution of the restored vision are still limited; current prosthetic devices cannot fully replicate the complexity and functionality of the natural retina;–Surgical complexity: Implanting retinal prostheses requires delicate and technically challenging surgical procedures with inherent risks;–Patient eligibility: Selection criteria are crucial to ensure that candidates have specific visual and anatomical characteristics that can benefit from the device; factors such as residual vision, retinal health, and overall eye condition need to be carefully evaluated, leading to a limited pool of eligible candidates;–Long-term durability: The longevity of retinal prostheses poses significant challenges; over time, the implant may encounter issues such as mechanical failure, degradation, or tissue response that can affect its performance, which are still unknown for now; we still do not know if a prostheses already implanted in a patient’s body can be repaired or replaced by a newer more developed version in the future;–Cost and accessibility: Retinal prostheses are currently expensive due to the advanced technology involved and the complexity of the surgical procedure; the high costs can restrict access to these treatments for many patients, limiting their availability and adoption in clinical practice; reducing costs and increasing accessibility are important considerations for the broader application of retinal prostheses.

Addressing these bottlenecks requires further advancements in technology, surgical techniques, and the understanding of retinal physiology.

The future prospects of retinal prostheses are dependent on improvements in clinical trial results that will then support the devices’ approval as well as increase their adoption amongst users. Bioelectronic implants such as the Argus II, Alpha IMS (Retina Implant AG, Reutlingen, Germany), BVT Bionic Eye System (Bionic Vision Technologies, Melbourne, Australia), and IMIE 256 were used in clinical trials [148]. Although PRIMA is currently under a clinical trial in patients with geographic atrophy, Argus II is the only retinal implant approved by the Food and Drug Administration, as it has restored some degree of vision in patients with advanced retinitis pigmentosa. However, Argus II implants hold a major disadvantage related to high stimulation that requires large electrodes, which results in either wide electrode spacing or low electrode density [79]. With these challenges, the IMIE 256 was recently upgraded to exhibit a smaller size and higher number of electrodes, which compensated for the limitations in the Argus II system. The IMIE 256 consists of an episcleral electronic implant body with an antenna for telemetry, 248 larger electrodes of 210 µm diameter, and 8 small electrodes of 160 µm diameter. As mentioned before, the IMIE 256 has been so far implanted in five patients with favorable outcomes in safety and clinical efficacy profiles. Additionally, results were reported regarding the risk of conjunctival breakdown and exposure of the episcleral electronic packaging being low with the IMIE 256, unlike with the Argus II system. The continuous iteration of such devices and their improvement will dictate whether retinal prostheses become widely adopted or replaced by other therapeutics.

Finally, the future outlook on retinal prostheses will be based on parallel growth in the field of artificial intelligence and the development of deep learning algorithms. Recently, there was a substantial breakthrough in the field of processing algorithms for retina prostheses, especially with the combination of the discovery of the retina’s working principle and state-of-the-art computer vision models [149]. This combination is resulting in remarkable progress and enhancement regarding the existing limitations in the field of retinal prostheses aimed to improve visual perception. AI-based image processing methods have a stronger extraction ability, providing greater convenience for patients with retinal diseases [150].

## 7. Conclusions

Overall, the field of retinal prosthetics has made significant strides in recent years, with advancements in wireless technology and artificial intelligence and the introduction of various implant designs. While challenges remain in improving visual resolution and image quality, minimizing surgical risks, and optimizing rehabilitation programs, the potential for retinal prostheses to restore vision in individuals with significant vision loss is promising.

The overview of different types of retinal prostheses, including epiretinal, subretinal, and suprachoroidal sensors, and the principles of electronic retinal prostheses, such as electrode size and material and charge density, were examined in detail. The clinical aspects of retinal prostheses, including safety and adverse events, visual function and outcomes, and rehabilitative programs, were also discussed.

While alternative therapies, including cell-based, gene-based, and optogenetic treatments, have gained prominence, retinal prostheses still maintain their significance in the management of retinal diseases. Future research focusing on the integration of various treatment modalities, including cell, gene, or optogenetics therapy, may potentially lead to enhanced vision restoration outcomes.

Collaboration across different disciplines, including bioengineering, electrical engineering, computer science, materials science, and ophthalmology, is critical to advancing the field of retinal prosthetics. Continued research and development in this area are vital for addressing the ongoing debate over the viability of implantable retinal devices and for bringing this life-changing technology to a larger population.

In conclusion, while retinal prostheses face challenges, their potential to restore vision and improve patients’ quality of life cannot be overlooked. With interdisciplinary collaboration, continued research and development, and a patient-centered approach, we can work toward achieving this goal and ultimately provide a life-changing solution for those who are affected.

## Figures and Tables

**Figure 1 sensors-23-05782-f001:**
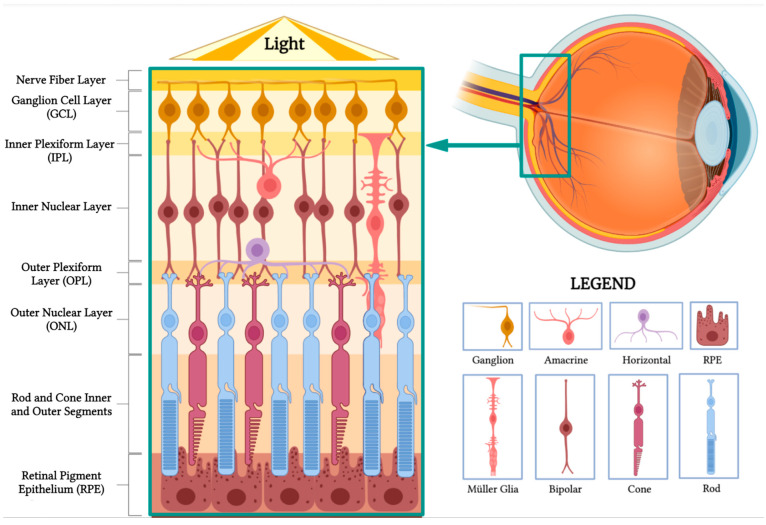
Retinal anatomy. The illustration highlights the different layers of the retina and its main cell types. (BioRender, https://app.biorender.com/, accessed on 16 February 2023).

**Figure 2 sensors-23-05782-f002:**
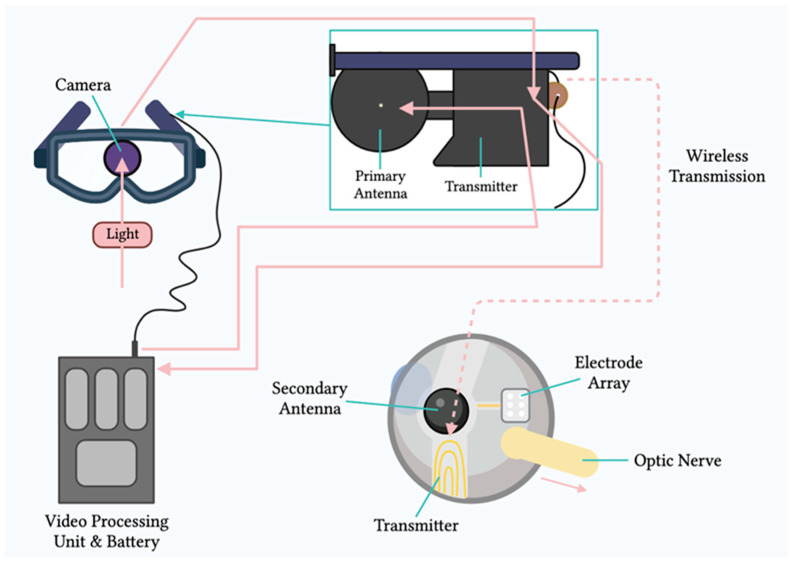
Components of the ARGUS II system. Schematic representation of the main components of the ARGUS II retinal prosthesis system, including the camera, video processing unit, and electrode array.

**Figure 3 sensors-23-05782-f003:**
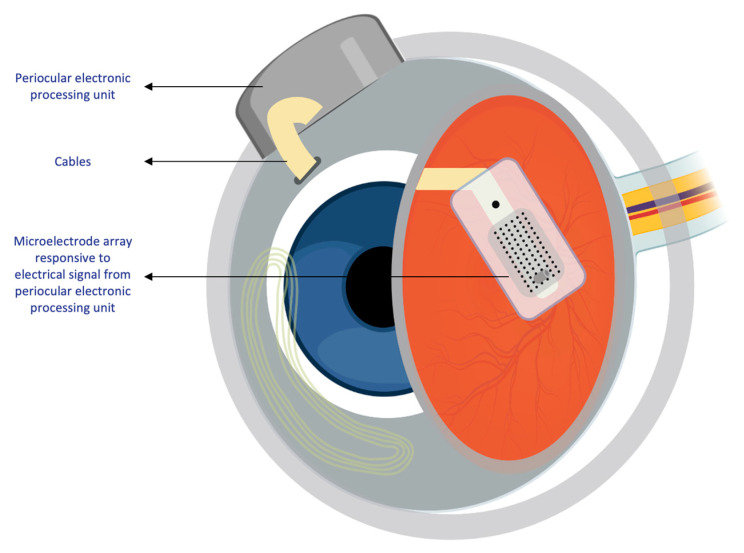
Intraocular components of the ARGUS II system. Diagram showing the internal components of the ARGUS II retinal prosthesis system that are implanted within the eye.

**Figure 4 sensors-23-05782-f004:**
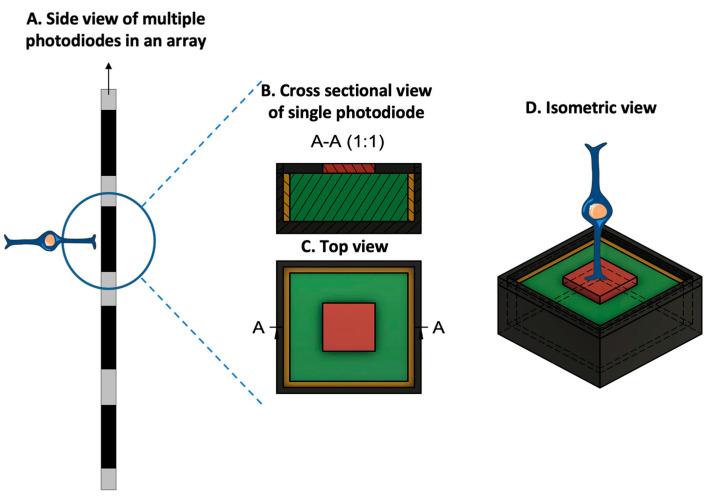
Prototypical design configuration of a photodiode, consisting of an inner electrode (red), a photodiode (green), an insulating layer (yellow), and an outer ground electrode (dark gray). (A. Side view of multiple photodiodes in an array. B. Cross-sectional view of the photodiode at position A-A, at the location specified in the top view. C. Top-orientation view of photodiode. D. Isometric view of the photodiode with a bipolar cell in close relation to the inner electrode. Due to the voltage drop between the inner and outer electrodes, the photodiode generates an electric field that can be used for cell stimulation. (Figure 4 was partly generated using Servier Medical Art, provided by Servier, licensed under a Creative Commons Attribution 3.0 Unported License.)

**Figure 5 sensors-23-05782-f005:**
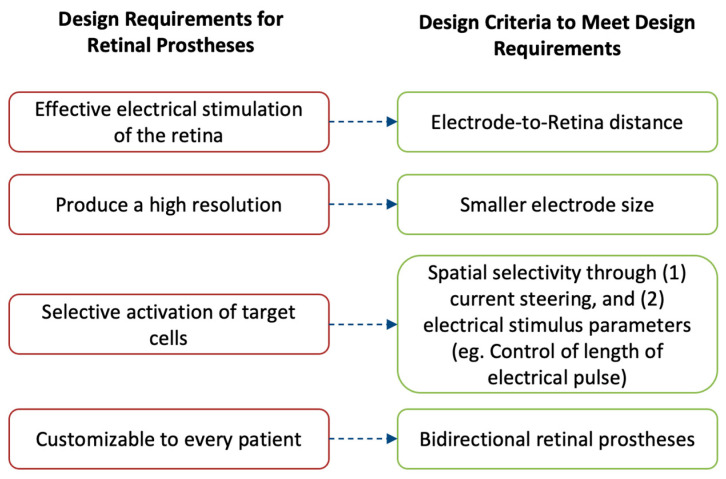
The performance of retinal prostheses is dependent on a few outcomes. Fulfilling these outcomes is important to drive their adoption in the market and by patients. These outcomes are (1) to provide effective electrical stimulation of the retina; (2) to produce a high-resolution image; (3) to selectively activate desired retinal cells, thereby avoiding image distortion; and (4) to be customizable for different patients. To achieve these requirements, retinal prostheses are designed with four main design criteria: (1) the electrode-to-retina distance, (2) having smaller electrode size, (3) implementing techniques to produce spatial selectivity, and (4) implementing bidirectional systems.

**Figure 6 sensors-23-05782-f006:**
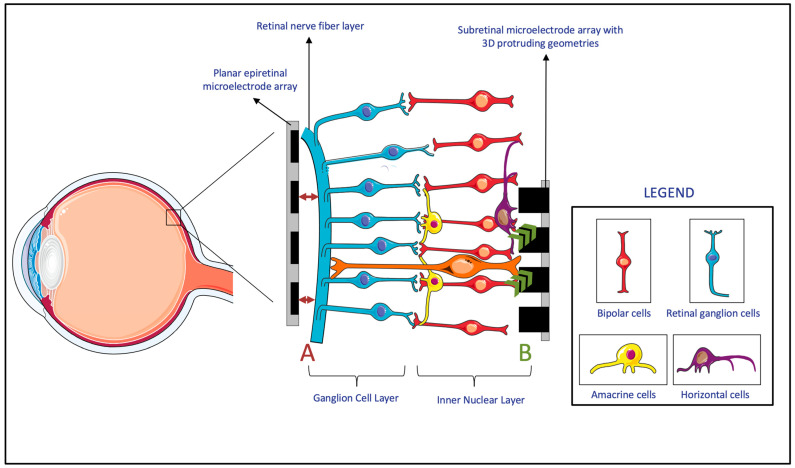
(A) The lack of the topographical alignment between a planar epiretinal microelectrode array and the retinal ganglion cells. (B) Migration and integration of the cells in the Inner Nuclear Layer to 3D protruding geometries in subretinal microelectrode array. (Figure 6 was partly generated using Servier Medical Art, provided by Servier, licensed under a Creative Commons Attribution 3.0 Unported License.)

**Figure 7 sensors-23-05782-f007:**
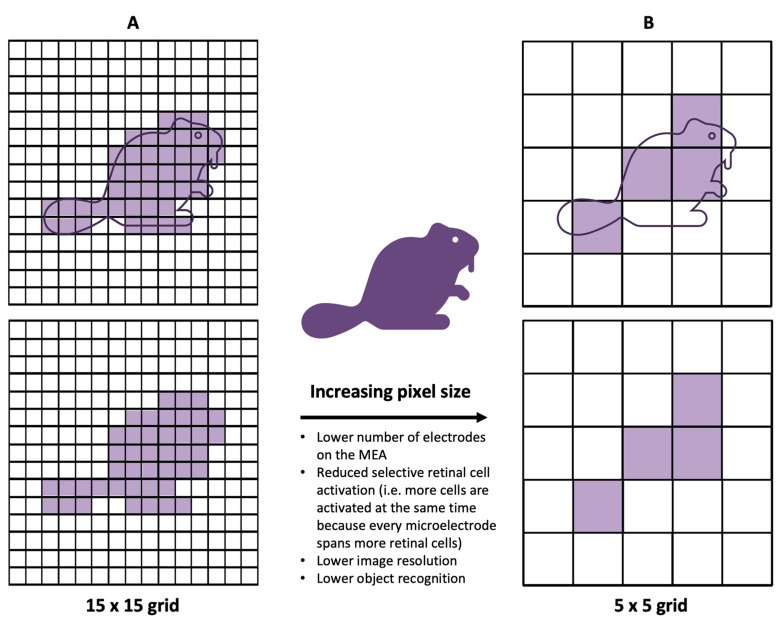
Demonstrating the impact of smaller electrodes. In comparison to the 15 × 15 grid (A), the 5 × 5 grid (B) produces a lower image resolution and enables less object recognition. Additionally, since each square in the grid represents the size of an electrode, having larger electrodes correlates to larger activation of retinal cells—i.e., less selective activation of desired, target retinal cells.

**Figure 8 sensors-23-05782-f008:**
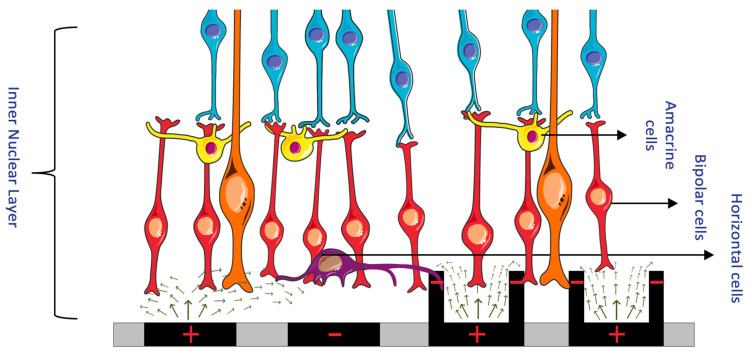
(Left) Electric field of a bipolar configuration causing lateral spread and unselectively stimulating many retinal cells. (Right) A 3D geometry electrode with circumferential returns generates locally confined electric fields, reducing electrode cross-talk and permitting more selective activation of retinal cells. (Figure 8 was partly generated using Servier Medical Art, provided by Servier, licensed under a Creative Commons Attribution 3.0 Unported License.)

**Figure 9 sensors-23-05782-f009:**
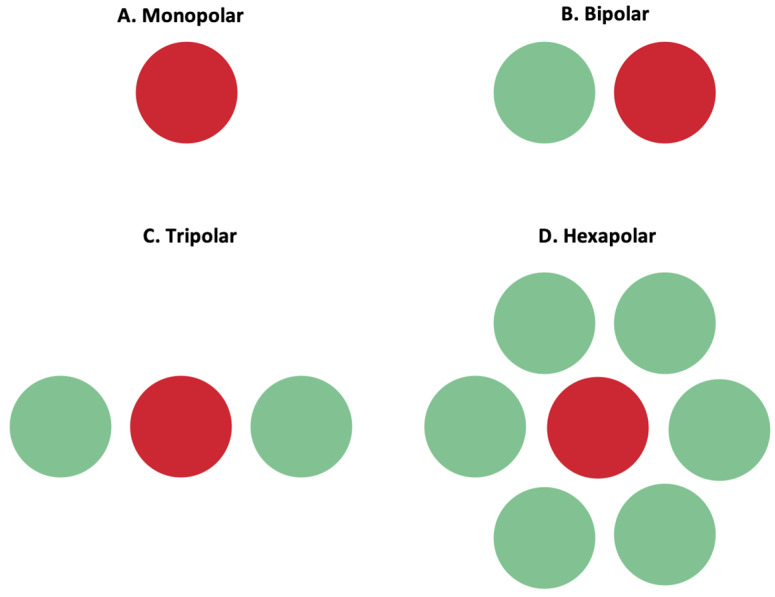
Active (red) and return (green) electrodes arranged in different configurations. (**A**) Monopolar configuration; (**B**) bipolar configuration; (**C**) tripolar configuration; (**D**) hexapolar configuration. These configurations impact the shape of the electric field, and the return electrode is used to limit current spread and produce a well-controlled, directed electric field.

**Table 1 sensors-23-05782-t001:** Adverse events, visual function, and outcomes of the different retinal prostheses.

Prosthesis Type		Adverse Events	Resolution of Adverse Events	Visual Function and Outcomes
Epiretinal	Argus II	Conjunctival erosions, hypotony, explantation, ocular inflammation, and retinal detachment.	Successfully treated or managed, except for hypotony and permanent retinal detachment.	Mixed visual function outcomes, self-report improvements in functional tasks.
	IRIS 2	Non-serious events: phlebitis, tack detachment, etc.; serious events: hypotony and persistent pain.	Successfully treated or managed.	Improved square localization, direction of motion detection, visual fields, and picture recognition with the device on.
Subretinal	Alpha IMS and AMS	Alpha IMS: increased intraocular pressure, retinal detachment, and retinal fibrotic changes; newer Alpha AMS: surgical dehiscence, implant displacement, partial silicone oil tamponade loss, and pain.	Successfully treated or managed.	Improved light source perception but difficulties with localizing and motion detection tasks; mixed benefits for daily living activities.
	PRIMA	Choroidal hemorrhage, subretinal hemorrhage, device displacement, and increased intraocular pressure due to medication non-adherence.	Successfully treated or managed.	Improved eccentric natural acuity and accurate identification of bar orientation.
Suprachoroidal		Fewer adverse outcomes compared to other types of implants; Non-serious events: pain, swelling, conjunctival injection, and local inflammation; one case of increased ocular pressure.	Successfully treated or managed.	Facilitated daily activities such as washing dishes, folding and organizing laundry, and identifying doorways and people in non-crowded spaces; difficulties remained in tasks such as identifying food on a plate; improved square localization and motion discrimination with the device on.

**Table 2 sensors-23-05782-t002:** Comparative analysis of emerging therapies for retinal diseases.

Treatment Modality	Description	Advantages	Limitations
Retinal Prostheses	This technology works by artificially stimulating the retinal nerve cells to mimic the function of lost or damaged photoreceptors.	Can restore some vision in patients with advanced retinal diseases, such as retinitis pigmentosa.	Safety and efficacy are still being evaluated. High cost may limit widespread adoption.
Cell-Based Therapies	Therapies involve the use of stem cells (pluripotent stem cells, bone marrow stem cells, and retinal progenitor cells) to replace or restore dysfunctional cells in the retina.	Potential to delay disease progression and restore vision loss, and can provide trophic support to remaining photoreceptors.	Potential risk of immune rejection.
Gene-Based Therapies	Therapies involve the use of viral and non-viral vectors and CRISPR-cas9 gene editing to correct genetic mutations causing retinal diseases.	Targets the root cause of the disease, potentially restoring vision.	Limited by the variety of gene mutations, so long-term outcomes and safety still require further investigation.
Optogenetics	This technique involves introducing photosensitive proteins to the degenerated retina to restore function and provide photosensitivity to remaining retinal cells.	Can restore photosensitivity to non-light-sensitive cells.	Requires further research on the structure, transport modes, dynamics, and optical properties of photosensitive proteins.

## Data Availability

Not applicable.

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
