# Peer review of "Retinal Prostheses: Engineering and Clinical Perspectives for Vision Restoration"

_sensors, 2023, doi:10.3390/s23135782_

Round 1

Reviewer 1 Report

Dear Authors,

the review reports what has been found in a few preclinical studies focused on retinal prostheses. in a review it would be appropriate to discuss the contents in a more consistent way, accompanying the preclinical studies with an important statistical analysis. The submissive work has a purely engineering cut, leaving only a small amount of space for the biological component.
it is also suggested that the authors eliminate the lanes from 62 to 65 as they contain redundant content with the above text.

Author Response

Dear Reviewer,

Thank you for your valuable feedback on our article. We appreciate your suggestions regarding the need for a more consistent and statistically supported discussion of the preclinical studies on retinal prostheses. We have carefully considered your comments and made revisions to address these concerns.

In response to your suggestion, we have included more numbers and statistics in the more clinical sections of the second part of our review article. These additions aim to enhance the robustness and credibility of the findings presented. We have also dedicated more space to the biological component in the second portion of our work, providing a deeper exploration of the relevant aspects with statistical support.

However, we would like to emphasize that the first part of our work, which focuses on explaining the functioning of retinal prostheses from an engineering perspective, plays a crucial role within the context of the scope of this journal and special issue. Our intention in starting with the engineering aspect before delving into the biological and clinical components is to cater to a diverse audience that may lack a background in biomedical engineering. By presenting the engineering foundation first, we aim to provide a comprehensive understanding of the subject matter for a wide range of readers.

That being said, we have taken your feedback into account and made further changes to the first part of our article. We have ensured that the engineering section is now more consistent, concise, and easier to read, aligning it with your suggestions. We believe these modifications will improve the overall flow and readability of the paper.

Once again, we appreciate your constructive feedback, which has helped us enhance the quality of our work. We are confident that the revised manuscript strikes a balance between the engineering, biological, and clinical components of the topic, making it accessible and informative for a broader readership.

Thank you for your time and consideration.

Reviewer 2 Report

I would like to thank the authors for this interesting review paper. Related work and results are clearly presented. However, there are still some problems to be solved, and the manuscript needs to be clarified in some places.

I have some suggestions:

1.     Page2 “1.2 Overview of retinal structure” the layers in text are different with layers in Figure 1.

2.     “Principles of Electronic Retinal Prostheses” is not easy to understand with the 3 figures. And Figure 3 and Figure 4 are not cited in the main text, please add it.

3.     In Figure 6, I’m confused by the left and right part, and the text direction is not comfortable for reading.

4.     The part of “Engineering of Retinal Prostheses” is structured from technical aspect, which is not easy to follow.

5.     In the part of “Clinical Considerations for Retinal Prostheses”, I suggest to use some tables for characteristics of the typical retinal prostheses’ types available to patients.

6.     In my opinion, despite approval by FDA, the effectiveness of Argus II treatment has not been confirmed. Retinal Implant Alpha AMS is approved by CE. Are Indications and Contraindications the same?

7.     Many therapeutic avenues are being explored for an increasing number of inherited retinal diseases, yet most of these diseases currently remain untreatable. What are the current bottlenecks limiting the clinical application of retinal prostheses?

8.     In the part of “Comparison of Alternative Emerging Therapies to Retinal Prostheses “, I also recommend to add some tables.

9.     The limitation of current prostheses can be discussed. From a security perspective of current prostheses, can the prostheses already implanted in the patient's body be repaired or replaced? When the technology you rely on is eliminated like old versions of consumer electronics, new problems will arise?

Author Response

Thank you for your insightful and constructive comments on our review paper. We sincerely appreciate your appreciation for the presented related work and results. We have carefully reviewed your suggestions and made the necessary revisions to address the issues you raised. We would like to respond to each of your suggestions individually:

  1. Page2 “1.2 Overview of retinal structure” the layers in text are different with layers in Figure 1.

Thank you for bringing the inconsistency between the text and Figure 1 to our attention in the section titled "1.2 Overview of retinal structure" on page 2. We appreciate your keen observation, and we have made the necessary changes to ensure consistency between the layers mentioned in the text and those depicted in Figure 1.

After carefully reviewing the section and the figure, we have revised both the text and the figure to accurately reflect the same layers of the retinal structure. The modifications were made to ensure that readers can easily follow the discussion and make appropriate connections between the information presented in the text and the visual representation in Figure 1.

  1. “Principles of Electronic Retinal Prostheses” is not easy to understand with the 3 figures. And Figure 3 and Figure 4 are not cited in the main text, please add it.

Thank you for the comment and your suggested amendments. We have edited the section to enhance legibility by expanding where appropriate, rephrasing elsewhere, and otherwise reconstructing the section. The figures, which were not previously cited in the text, are now included appropriately. Two of the three figures were updated to ensure that the wording in the main text directly corresponds with the labels in the figure, allowing the reader to correlate the main text with the figures.

  1. In Figure 6, I’m confused by the left and right part, and the text direction is not comfortable for reading.

Thank you again for your comment, we appreciate it. We have updated the figure to include label "A" and label "B" instead of "left" and "right". Moreover, we color coded label "A" and "B" to the specific details in the figure which we were aiming to highlight. Additionally, we added a legend to the figure which allowed us to make the text more comfortable to read.

  1. The part of “Engineering of Retinal Prostheses” is structured from technical aspect, which is not easy to follow.

Thank you for the feedback. To support the reader in understanding the section, we have added an introductory preamble which provides a big picture overview of four main performance outcomes of retinal prostheses. Each one of these outcomes is related to an engineering design criterion, which dictates the way that retinal prostheses are made. These design criteria are the four main subsections within "Engineering of Retinal Prostheses”. We have added a new figure to this preamble (figure 5) which summarizes the entire engineering section and provides the same overview in a figure format. Next, we edited all the subsections of the "Engineering of Retinal Prostheses" section, breaking down concepts further, ensuring that they are clearer to the first-time reader. Finally, seeing that some concepts were still complex, we also added another figure (figure 7) which should help the reader visualize a concept that is difficult to convey in just words.

  1. In the part of “Clinical Considerations for Retinal Prostheses”, I suggest to use some tables for characteristics of the typical retinal prostheses’ types available to patients.

We appreciate your insightful suggestion regarding the inclusion of a table summarizing the characteristics of typical retinal prostheses available to patients in the "Clinical Considerations for Retinal Prostheses" section. We agree that such a table would provide valuable information and enhance the understanding of readers, particularly patients.

Taking your suggestion into account, we have incorporated a table that summarizes the relevant characteristics of retinal prostheses. In this table, we have included important details related to adverse events and their management, as well as the visual outcomes associated with these prostheses. By presenting this information in a tabular format, we aim to provide a concise and easily accessible overview for readers, allowing them to quickly grasp the essential aspects of each type of retinal prosthesis.

We would like to express our gratitude for bringing this idea to our attention, as we believe the addition of this table significantly enhances the clarity and usefulness of the "Clinical Considerations for Retinal Prostheses" section.

Prosthesis Type

Adverse events

Resolution of adverse events

Visual Function and Outcomes

Epiretinal

Argus II

Conjunctival erosions, hypotony, explantation, ocular inflammation, and retinal detachment.

Successfully treated or managed, except for hypotony and permanent retinal detachment.

Mixed visual function outcomes, self-report improvements in functional tasks.

IRIS 2

Non-serious events: phlebitis, tack detachment, etc.; Serious events: hypotony and persistent pain

Successfully treated or managed

Improved square localization, direction of motion detection, visual fields, and picture recognition with the device on.

Subretinal

Alpha IMS and AMS

Alpha IMS: Increased intraocular pressure, retinal detachment, and retinal fibrotic changes. Newer edition Alpha AMS: surgical dehiscence, implant displacement, partial silicone oil tamponade loss, and pain.

Successfully treated or managed

Improved light source perception but difficulties in localizing and motion detection tasks. Mixed benefits in daily living activities.

PRIMA

Choroidal hemorrhage, subretinal hemorrhage, device displacement, and increased intraocular pressure due to medication non-adherence.

Successfully treated or managed

Improved eccentric natural acuity and accurate identification of bar orientation.

Suprachoroidal

Fewer adverse outcomes compared to other types of implants. Non-serious events: pain, swelling, conjunctival injection, and local inflammation. One case of increased ocular pressure.

Successfully treated or managed

Facilitated daily activities such as washing dishes, folding and organizing laundry, and identifying doorways and people in non-crowded spaces. Difficulties remained in tasks like identifying food on a plate. Improved square localization and motion discrimination with the device on.

  1. In my opinion, despite approval by FDA, the effectiveness of Argus II treatment has not been confirmed. Retinal Implant Alpha AMS is approved by CE. Are Indications and Contraindications the same?

Thank you for your valuable feedback. We appreciate your suggestion, and we agree that it is important to acknowledge the concerns about the effectiveness of Argus II treatment. In response to your comment, we have included the following sentence: "The Argus II is currently the only FDA-approved retinal prosthesis device in North America. However, despite FDA approval, its effectiveness has not been confirmed." We believe this addition helps provide a balanced perspective on the current status of Argus II treatment. Thank you for bringing this to our attention, and we strive to address all relevant viewpoints in our content.

Regarding the comparison between the indications and contraindications of Argus II and Alpha AMS, we have provided additional information in the "Importance of Patient Selection and Screening" section. In this section, we have discussed the indications and contraindications of both devices, highlighting any similarities and differences between them. More specifically, we’ve added the following statement with new references:

“The indications and contraindications of the retinal prosthesis Alpha AMS are similar to Argus II, with the exception that there is no age specification mentioned in the literature for Alpha AMS implants. However, it is important to note that for Alpha AMS, the retina must have a thickness of > 100 μm to require functionality. More precisely, the indications for Alpha AMS implantation include [70]:

  • Light perception without projection or no light perception in hereditary retinal diseases (Retinitis Pigmentosa, Choroideremia, Usher Syndrome).
  • Primary rod cone or cone rod dystrophies in their end stage diseases.
  • Have a prior history of normal visual function for > 12 years.
  • Prior history of Pseudophakia or aphakic status prior to retinal prosthesis implantation
  • Fluorescein angiography showing retinal vascular perfusion in all four quadrants of macula.
  • Evidence of inner retinal function (ganglion cells and optic nerve function) observed through the ability to elicit phosphene thresholds.
  • Patient must be able to give written informed consent and to attend follow-up and visual rehabilitation.

Contraindications for Alpha AMS implantation include [70,83]:

  • Ophthalmic conditions with relevant effects upon visual function (Glaucoma, diabetic neuropathy, retinal detachment, optic neuropathies, heavy clumped pigmentation at pos-terior lobe, cystoid macular edema)
  • Retina < 100 μm or no layering of the inner retina shown by OCT.
  • Scar tissue (epiretinal, intraretinal, subretinal, macular pucker)
  • Occipital stroke
  • Congenital blindness, severe amblyopia
  • Substantial corneal opacity or any opacification of ocular structures that prevent clear image transmission.
  • Active inflammation (Uveitis)
  • Systemic conditions that could pose significant risks during general anesthesia (Cardi-ovascular/ pulmonary/ severe metabolic conditions like diabetes)
  • Life expectance < 1 year
  • Inability to comply with post-operative follow-up and rehabilitation due to psychiatric/ Neurological diseases (Parkinson, dementia, MS, epilepsy, severe depression and anxie-ty)

We hope that the information provided in the "Importance of Patient Selection and Screening" section addresses your concerns and provides a comprehensive understanding of the indications and contraindications for both Argus II and Alpha AMS.

  1. Many therapeutic avenues are being explored for an increasing number of inherited retinal diseases, yet most of these diseases currently remain untreatable. What are the current bottlenecks limiting the clinical application of retinal prostheses?

We thank the reviewer for this appealing question. Main bottlenecks limiting the application of retinal prosthesis in clinical practice are mainly limited effectiveness, surgical complexity, limited eligibility criteria, unknown long-term durability, cost, and accessibility. A new paragraph discussing the numerous bottlenecks limiting the application of retinal prosthesis was added successfully:

“The clinical application of retinal prostheses faces several bottlenecks that limit their widespread use. These include:

-     Limited effectiveness: While retinal prostheses can provide some degree of visual perception, the quality and resolution of the restored vision are still limited. Current prosthetic devices cannot fully replicate the complexity and functionality of the natural retina.

-     Surgical complexity: Implanting retinal prostheses requires delicate and technically challenging surgical procedures with inherent risks.

-     Patient eligibility: Selection criteria are crucial to ensure that candidates have specific visual and anatomical characteristics that can benefit from the device. Factors such as residual vision, retinal health, and overall eye condition need to be carefully evaluated, leading to a limited pool of eligible candidates.

-     Long-term durability: The longevity of retinal prostheses poses significant challenges. Over time, the implant may encounter issues such as mechanical failure, degradation, or tissue response that can affect its performance, which are still unknown for now. We still do not know if the prostheses already implanted in the patient's body can be repaired or replaced by a newer more developed version in the future.

-     Cost and accessibility: Retinal prostheses are currently expensive due to the advanced technology involved and the complexity of the surgical procedure. The high costs can restrict access to these treatments for many patients, limiting their availability and adoption in clinical practice. Reducing costs and increasing accessibility are important considerations for the broader application of retinal prostheses.

Addressing these bottlenecks requires further advancements in technology, surgical techniques, and understanding of retinal physiology.”

  1. In the part of “Comparison of Alternative Emerging Therapies to Retinal Prostheses “, I also recommend to add some tables.

We appreciate your suggestion regarding the inclusion of table in the "Comparison of Alternative Emerging Therapies to Retinal Prostheses" section. We agree that tables can be an effective way to present comparative information in a concise and organized manner.

In response to your recommendation, we have now added Table 2 to the manuscript. This table provides a cocise and brief comparison of the alternative emerging therapies to retinal prostheses that were discussed in the section. It presents key characteristics, advantages, and limitations of each therapy, allowing readers to easily grasp the similarities and differences between them.

By incorporating this table, we aim to enhance the clarity and accessibility of the information provided in the "Comparison of Alternative Emerging Therapies to Retinal Prostheses" section.

  1. The limitation of current prostheses can be discussed. From a security perspective of current prostheses, can the prostheses already implanted in the patient's body be repaired or replaced? When the technology you rely on is eliminated like old versions of consumer electronics, new problems will arise?

Thank you for bringing up this important point. We acknowledge that the long-term reliability and durability of retinal prostheses are critical challenges that still lack definitive answers. As mentioned earlier in our response to remark #7, we have extensively addressed these bottlenecks and uncertainties in the corresponding section.

In that section, we have highlighted the key concerns associated with the long-term reliability and durability of retinal prostheses. We have discussed the current limitations and ongoing research efforts aimed at addressing these challenges. By delving into these bottlenecks, we have emphasized the complexity of the issue and the need for further investigation.

We appreciate your feedback, which has allowed us to ensure a better flow and coherence within our manuscript.

Reviewer 3 Report

Review manuscript "Retinal Prostheses: Engineering and Clinical Perspectives for  Vision Restoration" by Wu is interesting work on very difficult problem. Authors step by step describes current situation on development and application of such unique prostheses for human eye. On my mind, all aspects are mentioned and discussed. Actually authors reviewed even larger period of investigations as mentioned. It is huge amount of information which is presented in review. I suggest to add

1. Layout of review

2. Abbreviation list

Author Response

Thank you for your valuable feedback on our manuscript "Retinal Prostheses: Engineering and Clinical Perspectives for Vision Restoration." We appreciate your positive remarks and are glad that you found our work interesting. We have carefully considered your suggestions and have made the necessary revisions accordingly.

Regarding your first suggestion, we have included a layout of the review as Figure 1 in the revised version of the manuscript. This figure provides a visual representation of the organization and flow of the review, allowing readers to quickly grasp the overall structure.

Additionally, we have addressed your second suggestion by ensuring that all abbreviations used in the manuscript are accompanied by their full name upon first appearance. We understand the importance of clarity and adherence to the journal's requirements. Furthermore, we have included an abbreviation list at the end of the manuscript to provide readers with a convenient reference.

We believe that these additions and modifications have improved the overall clarity and readability of the manuscript. Thank you once again for your valuable input, which has helped us enhance the quality of our work.

Round 2

Reviewer 1 Report

Dear Authors,

Thank you for accepting the revisions and suggestions. I believe that the changes made make your paper transversal between the biotechnological and engineering fields

Reviewer 2 Report

The authors have well addressed all my comments. Thanks for an interesting paper.